# Achilles' Heel of Mamba: Essential difficulties of the Mamba architecture demonstrated by synthetic data

**Tianyi Chen**[1,2]*, **Pengxiao Lin**[1,2]*, **Zhiwei Wang**[1,2], **Zhi-Qin John Xu**[1,2,3]†
[1] Institute of Natural Sciences, MOE-LSC, Shanghai Jiao Tong University
[2] School of Mathematical Sciences, Shanghai Jiao Tong University
[3] Shanghai Seres Information Technology Co., Ltd, Shanghai 200040, China

## Abstract

State Space Models (SSMs) have emerged as promising alternatives to attention mechanisms, with the Mamba architecture demonstrating impressive performance and linear complexity for processing long sequences. However, the fundamental differences between Mamba and Transformer architectures remain incompletely understood. In this work, we use carefully designed synthetic tasks to reveal Mamba's inherent limitations. Through experiments, we identify that Mamba's nonlinear convolution introduces an asymmetry bias that significantly impairs its ability to recognize symmetrical patterns and relationships. Using composite function and inverse sequence matching tasks, we demonstrate that Mamba strongly favors compositional solutions over symmetrical ones and struggles with tasks requiring the matching of reversed sequences. We show these limitations stem not from the SSM module itself but from the nonlinear convolution preceding it, which fuses token information asymmetrically. These insights provide a new understanding of Mamba's constraints and suggest concrete architectural improvements for future sequence models.

## 1 Introduction

Large Language Models (LLMs) have achieved remarkable progress and are now widely applied across a broad range of fields (Vaswani et al., 2017; Liu et al., 2018; Devlin et al., 2018; Radford et al., 2019; Touvron et al., 2023; OpenAI, 2023; Brown et al., 2020; Dong et al., 2022; Garg et al., 2022; Trinh et al., 2024; Davies et al., 2021). The performance and inductive biases of such models are largely determined by their underlying architectures. Among these, the Transformer (Vaswani et al., 2017) has become a dominant backbone in LLMs. Its attention mechanism is central to its strong performance (Brown et al., 2020; Olsson et al., 2022; Wang et al., 2024). However, this mechanism also represents one of its major limitations: the computational complexity of attention scales quadratically with sequence length, making Transformers inefficient for long-sequence tasks.

To address this issue, various Transformer variants have been proposed to reduce the computational cost of attention, including sparse attention (Child et al., 2019) and linear attention mechanisms (Katharopoulos et al., 2020). Mamba (Gu and Dao, 2024; Dao and Gu, 2024), which incorporates State Space Model (SSM), has recently garnered significant attention due to its linear complexity with respect to sequence length and its superior performance on long-sequence problems. As a result, SSM offering a natural, computation-efficient alternative similar to linear attention have become a focal point of research.

---

*Equal contribution.
†Corresponding author: xuzhiqin@sjtu.edu.cn.

39th Conference on Neural Information Processing Systems (NeurIPS 2025).

To better understand the behavior of large-scale models and to guide meaningful architectural improvements, it is crucial to investigate the underlying causes of the differences between Mamba and Transformer. Given the inherent complexity of natural language tasks, this study leverages simple but meticulously crafted synthetic data.

When examining architectural details more closely, is the difference between Mamba and Transformer limited merely to the replacement of attention with State Space Models (SSMs)? In fact, beyond the use of SSMs, Mamba exhibits several fundamental differences from the Transformer architecture. One of the most critical distinctions lies in Mamba's use of nonlinear convolution (O'shea and Nash, 2015). On one hand, the nonlinear convolution enables Mamba to propagate information within a sequence without relying solely on the SSM unlike Transformer, where Transformer depends entirely on attention for intra-sequence communication. On the other hand, this convolution fuses token-level information within the sequence, and the SSM operates on this fused representation to perform matching and extraction.

Notably, the nonlinear convolution in Mamba introduces an intrinsic asymmetry due to the asymmetric structure of convolution kernels. This asymmetry is transferred to the fused token representations and consequently affects how information is extracted. To better understand this limitation, we devised a composite function task (Zhang et al., 2024a,b, 2025) aimed at evaluating how Mamba handles compositional structures.

The composite function task admits two possible solutions: a composite solution and a symmetric solution. We observe that Mamba exhibits a strong bias toward the composite solution while struggling to learn the symmetric one. We find that Mamba struggles to match sequences under order changes–for example, "1234" vs. "4321". To test this limitation, we designed a inverse sequence matching task, where the model must match a sequence with its reversed counterpart. Experimental results confirm that Mamba has difficulty completing this task, whereas Transformer handles it with ease. We further introduced a residual connection, complemented by positional encoding, allowing the SSM to directly handle data without relying on convolution. This modification led to a significant improvement in Mambas performance on the inverse sequence matching task, confirming that the core issue lies not within the SSM itself, but rather in the nonlinear convolution outside the SSM. This work therefore provides a new angle of symmetry to understand fundamental mechanisms of Mamba structure.

The main contributions of this work are as follows:

1. We conduct an in-depth analysis of how Mamba solves the composite function task, revealing a fundamental difference in information acquisition between Mamba and Transformer. We show that Mamba tends to rely on convolution to retrieve relevant information.

2. Through systematic experiments and observations, we identify Mamba's bias toward asymmetric solutions in the composite function task. We demonstrate that this behavior distinguishes Mamba from Transformer and trace the root cause to the asymmetry introduced by convolution.

3. To further examine this asymmetry bias, we design an inverse sequence matching task, in which Mamba exhibits clear difficulties. We show that these difficulties can be effectively addressed through targeted modifications inspired by our earlier findings, leading to substantial performance improvements.

4. By highlighting how nonlinear convolution-based fusion induces an inherent asymmetry in Mamba, we provide insightful guidance for future model improvements and the design of new architectures.

## 2   Related work

**State Space Models (SSM), Mamba, and Their Shortcomings.** State space models (SSM) originate from neuromorphic spiking models (Voelker, 2019; Voelker et al., 2019) and have gained prominence through several key developments, such as S4 (Gu et al., 2022), S5 (Smith et al., 2023), the RWKV series (Peng et al., 2023, 2024), RetNet (Sun et al., 2023), and GLA (Yang et al., 2024). Among these, Mamba2 (Gu and Dao, 2024) stands out by delivering performance competitive with Transformers while requiring significantly lower computational resources. However, numerous studies reveal that Mamba has notable limitations. Research including (Ben-Kish et al., 2025; Ye et al.,

2025; Yuan et al., 2025) indicates that Mamba generally underperforms compared to Transformers in tasks involving long-context understanding, prompting the development of alternative models like DeciMamba, LongMamba, and ReMamba. Similarly, (Park et al., 2024; Waleffe et al., 2024) explore Mambas in-context learning abilities and conclude that they fall short of Transformer capabilities. Additionally, (Arora et al., 2023; Jelassi et al., 2024) demonstrate that Mamba struggles with retrieval tasks, such as copying information from the input context. Some studies, such as (Xu et al., 2025), also examine the practical efficiency of Mamba compared to Transformers, noting that despite its theoretical advantages, real-world performance can be lacking. Furthermore, (Ren et al., 2024) introduces a COPY task, which exposes a performance bottleneck in Mamba. Our study takes a structural view of Mamba, employing carefully designed synthetic data to explore its behavior from the perspective of symmetry, offering deeper insights into these shortcomings of Mamba and proposing a solution to address them.

**Understanding the Mechanism of Neural Network Models.** Our work conducts an in-depth investigation into the internal mechanisms of Mamba. In identifying the functional roles of key modules in Mamba presented in this paper, we adopt the commonly used techniques of perturbation and causal intervention similar to research on interpretability in large language models (Vig et al., 2020; Jeoung and Diesner, 2022; Wang et al., 2022; Conmy et al., 2023; Merullo et al., 2023; Guo et al., 2023; Wang and Weinan, 2024; Amsel et al., 2024; Li et al., 2024; Wang et al., 2024). The construction of synthetic datasets in our study is primarily inspired by the works of Poli et al. (2024) and Zhang et al. (2024a). The design of our composite function tasks is adopted from the approach used in Zhang et al. (2024b,a). Additionally, several insightful theoretical studies on feedforward neural networks have also informed our theoretical analysis. For instance, various works explore the preferences and generalization capabilities of neural networks from perspectives such as regularization, frequency, and dynamics (Xu et al., 2025a,b, 2019; Wang et al., 2024; Jacot et al., 2018, 2020; Arora et al., 2019, 2018; Wu and Su, 2023; Wang and Wu, 2023; Arora et al., 2022; Li et al., 2021; Wu et al., 2018; Zhu et al., 2018; Arora et al., 2019; Ren et al., 2024).

## 3 Preliminary

### 3.1 Introduction of Mamba

The overall structure of Mamba (Gu and Dao, 2024; Dao and Gu, 2024) is shown in Fig. 1. The Mamba block can be divided into three parts: the widely known SSM component, the pre-SSM part, and the post-SSM part. Omitting trivial dimension transformations and setting the batch size to 1 to omit the batch dimension, for a given input $U$ to the block, the internal computation process to obtain the output $O$ can be described as follows:

**Pre-SSM**

$$(\tilde{U}, Z, dt) = \textbf{Linear}(U), \quad U \in R^{(s,d)}, \tilde{U} \in R^{(s,2d+2h)}, Z \in R^{(s,2d)}, dt \in R^{(s,N_h)}, \quad (1)$$

$$(B, C, X) = \boldsymbol{\sigma}(\textbf{Conv1d}(\tilde{U})), \quad B \in R^{(s,h)}, C \in R^{(s,h)}, X \in R^{(s,2d)}, \quad (2)$$

$$\tilde{X} = X \circ dt, X \in R^{(N_h, s, 2d/N_h)}, \quad (3)$$

**SSM**

$$Mask = \textbf{F}(A_0, dt), \quad A_0 \in R^{N_h}, Mask \in R^{(N_h, s, s)}, \quad (4)$$

$$I = \textbf{Repeat}(CB^\top, N_h)), \quad I \in R^{(N_h, s, s)}, \quad (5)$$

$$S = Mask \circ I, \quad S \in R^{(N_h, s, s)}, \quad (6)$$

$$Y = S\tilde{X} + X, \quad Y \in R^{(N_h, s, 2d/N_h)}, \quad (7)$$

**Post-SSM**

$$Y_{Norm} = \textbf{RMS}(Y \circ (\boldsymbol{\sigma}(Z))), \quad Y_{Norm} \in R^{(s,2d)}, \quad (8)$$

$$O = \textbf{Linear}(Y_{Norm}), \quad O \in R^{(s,2d)}, \quad (9)$$

where $s$ is the sequence length, $d$ is the model's hiddenstate dimension, $h$ is the SSM hidden dimension, $S$ is the SSM matrix, $N_h$ is the number of SSM heads, **Linear** denotes a linear transformation,

**Conv1d** denotes a one-dimensional convolution, $\sigma$ denotes a nonlinear activation function, **F** denotes the function that generates the $Mask$, **Repeat** denotes a dimension replication operation, $\circ$ denotes pointwise multiplication, and **RMS** denotes RMS normalization.

We define the composition of $\sigma$ and **Conv1d** as **nonlinear convolution** and **attention score** from token $j$ to token $i$ is given by the $(i, j)$ entry of the SSM matrix $S$ throughout the paper. For more comprehensive computational details, please refer to the appendix.

## 3.2 Difference between Mamba and Transformer

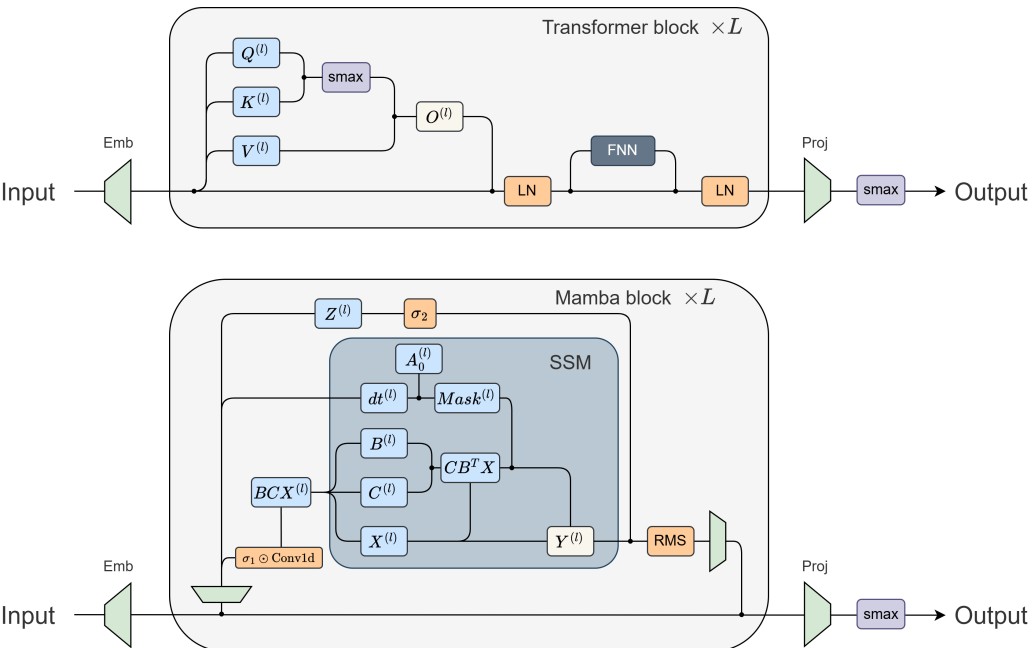

Figure 1: Overview of Mamba and Transformer Blocks. The green trapezoids represent linear mappings. "smax" denotes the softmax function, "FNN" stands for feed-forward neural network, and "LN" represents layer normalization. The meanings of variables specific to the Mamba block are explained in the main text.

By incorporating the SSM module, Mamba circumvents the quadratic complexity of attention in Transformers. While SSM significantly improves computational efficiency, the key difference from attention appears limited to the softmax function and the learnable mask. However, notable distinctions also exist beyond the SSM itself.

Prior to the attention-like SSM, Mamba applies a nonlinear convolution that fuses information, limiting the SSM to operate on already mixed representations. Mamba contains a z-gate following SSM, lacks a Feed-Forward Network (FFN), and its preceding nonlinear layer lacks hidden layers.

As we will demonstrate, nonlinear convolution is a defining feature of Mamba that underlies its fundamental divergence from the Transformer.

## 3.3 Composite function task

A detailed illustration of the composite function task (Zhang et al., 2024a,b, 2025) is shown in Fig. 2. The core idea of the composite function task can be understood with a simple real world analogy. Imagine a row of people indexed by numbers. Let's define two functions: $n(x)$: Find the person n positions to the right of person $x$ and $m(x)$: Find the person m positions to the left of person $x$. The composite function $m(n(x))$ means "If you start with person $x$, go n people to their right, and then m people to their left, who do you end up with?"

In the composite function task, each sequence contains two anchors and one key, the remaining elements in the sequence are randomly sampled from the same range as the key. Each anchor is

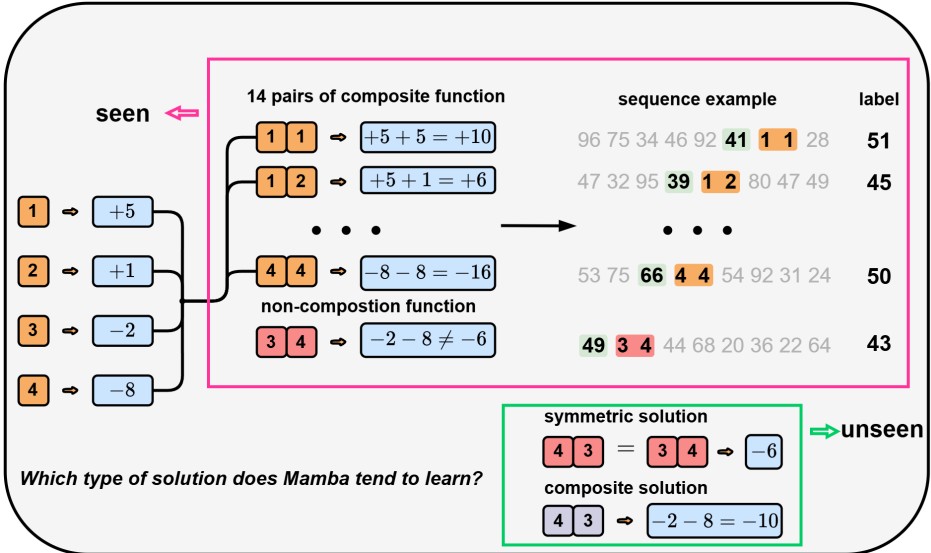

Figure 2: Illustration of the Composite Function Task. Anchors 1, 2, 3, and 4 (depicted in orange) each represent distinct functions. Among the 16 anchor pairs formed, 14 correspond to composite functions derived directly from the sequential application of the individual anchor functions. The pair "34", highlighted in red, is defined as a different function rather than a direct composition. The pair "43" is intentionally excluded from the training set. The input to each anchor pair function is referred to as the "key" (indicated in green). Label indicates the output of an anchor pair applied to a key.

assigned a unique function, and a pair of anchors defines a composite function, with the key (token before anchor pairs) serving as the input. We use the token 1 through 4 as anchors, yielding a total of 16 possible anchor pairs. Among these, 14 composite functions are defined by directly composing the functions associated with the individual anchors. For the anchor pair "34", however, we assign a function that deviates from the standard composition of its components.

The objective is to examine how Mamba handles the unseen anchor pair "43". Two plausible solutions exist: one is the **symmetric solution**, which infers the result of "43" by symmetry from the result of "34"; the other is the **composite solution**, which computes the result by directly composing the functions associated with "4" and "3".

In all experiments, the loss is calculated exclusively based on the final token of the sequence and its corresponding label. For detailed data and training settings, please refer to the appendix.

## 3.4 Inverse sequence matching task

To investigate whether Mamba can match token information fused by the nonlinear convolution, we design an inverse sequence matching task. The detailed structure of the task is illustrated in Fig. 3. The inverse sequence matching task corresponds to real world situations where people search for the symmetrical counterpart of an object or text. It is similar to asking children to find the symmetrical counterpart of a toy and fit them together. This task can test a model's ability to perform symmetry matching. Although it is simple and fundamental, it reflects a core capability of the model.

Each sample is built from a generating set of three distinct numbers (e.g., $\{28, 92, 37\}$). From its six possible permutations, five are randomly chosen to form key sequences, which are concatenated and separated by a token not in the generating set.

Next, one of the five key sequences is randomly selected, reversed, and appended to the end as the query sequence, separated by non-generating-set tokens so that the query and key sequences would not fuse by convolution. The number of tokens is determined by Mamba's pure convolutional receptive field. For example, in the illustration, six tokens correspond to the pure convolutional receptive field (refers exclusively to how many tokens can be accessed through convolution alone,

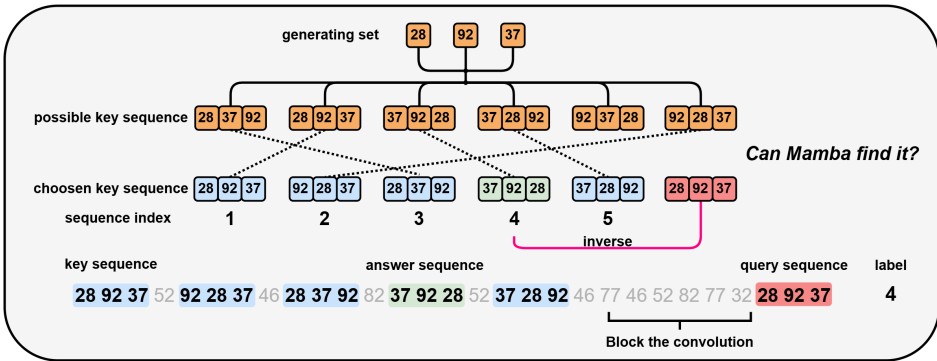

Figure 3: Illustration of the Inverse Sequence Matching Task. The orange elements denote the generation set, which consists of three distinct numbers randomly selected from the interval [20, 100], as well as all possible permutations thereof. Blue and green indicate selected key sequences from the permutation space, separated by random numbers that do not belong to the generation set. One green key sequence is chosen as the answer sequence. The query sequence (shown in red) is obtained by reversing the answer sequence. The corresponding label identifies the position of the answer sequence. To prevent Mamba from leveraging its nonlinear convolution mechanism to infer answers, we prepend the query sequence with random numbers (outside the generation set) matching the length of Mamba's pure convolutional receptive field.

without invoking the SSM) of a two-layer Mamba ($2 \times 3 = 6$). Finally, the label is the position index of the answer sequence, i.e., the unreversed version of the query sequence.

## 4    Mamba biases composite solution

In this section, we will empirically show that Mamba is struggling with the symmetric property in composition but biases composite solutions, and analyze the role of nonlinear convolution.

Initialization scale is shown to be critical for a Transformer to learn the composite or symmetric solution (Zhang et al., 2024b, 2025). Therefore, we scan different initialization scales as follows. A parameter $W \in \mathbb{R}^{d_1 \times d_2}$ is initialized as a Gaussian distribution $\mathcal{N}(0, (1/d_1^\gamma)^2)$, where $\gamma$ is called initialization rate (Luo et al., 2021).

We scan a Mamba with different layers and with different $\gamma$. As shown in Fig. 4, where the accuracy is computed by regarding the label as the composite solution in Fig. 4a or the symmetric solution in Fig. 4b, for small $\gamma$ (large initialization), the network fails to capture either composite or symmetric solution. The value at each position in the figure is computed as the average of three independent random runs. This is consistent with previous study in Transformer (Zhang et al., 2024b, 2025). For large $\gamma$ (small initialization), the network biases the composite function across almost all cases.

### 4.1    SSM does not function in composite tasks

It requires information from anchors and key to finish a composition task. Mamba has two potential options: utilizing the convolution or the SSM module. We found that in standard Mamba (Dao and Gu, 2024), convolution plays a critical role while SSM module does not function.

**Information blocking.**    The information flow analysis is a useful tool to visualize the information exchange on token level (Wang et al., 2024). Since the SSM module has high similarity with the attention, we treat every element in SSM matrix as the "attention score" in information flow in Fig. 5. The result suggests that, within the SSM, tokens at later positions have little attention to the key and anchor information. In the case of the Transformer, failure to retrieve the required information via attention renders it incapable of solving the composite function task. To further verify that Mamba does not utilize the SSM for information propagation, we applied a causal intervention approach (Feng and Steinhardt, 2023; Meng et al., 2022; Vig et al., 2020; Wang et al., 2024), manually blocking all information flow from the key and both anchors to the downstream tokens. The $(i, j)$ element of $S$ represents how much token $i$ attends to token $j$. If this value is set to 0, it implies that token

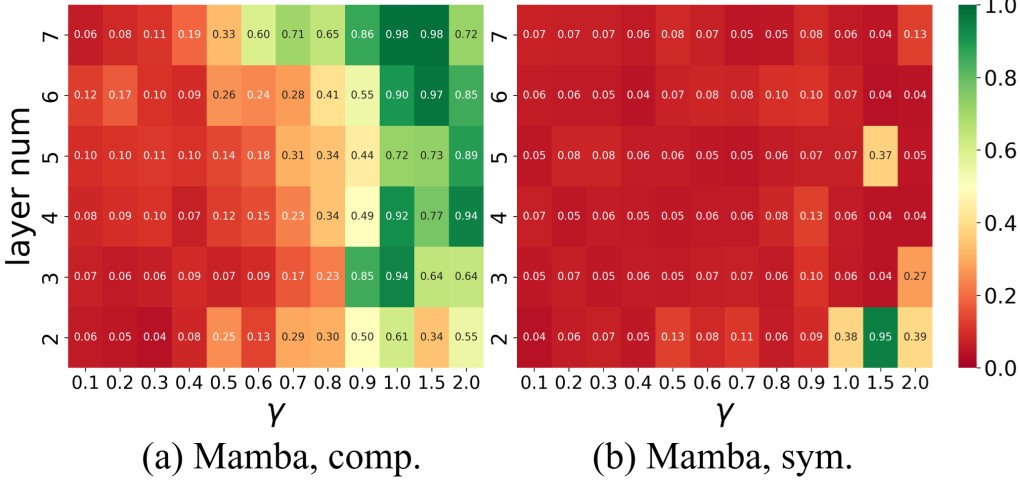

Figure 4: Phase diagram of Mamba on the composite function task. Accuracy (color) for composite function task under different initialization rates (abscissa) and depths (ordinate). The groundtruth for (a) is composite solution and for (b) is symmetric solution. Detailed model configurations and training settings are provided in the appendix.

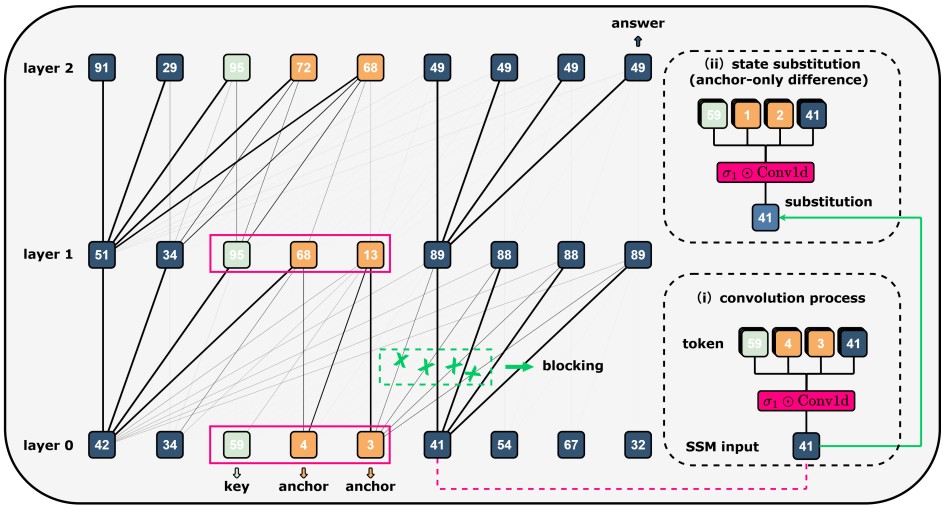

Figure 5: SSM information flow in Mamba for the composite function task. Left: SSM information flow; green crosses indicate pruned connections. Right: (i) SSM input computation; (ii) state replacement after convolution. Flow is computed from $S = Mask \circ C^T B$, with line thickness indicating flow magnitude. Attention score from token $j$ to token $i$ is given by the $(i, j)$ entry of $S$ The numbers are the outputs of each layer through the models final linear layer and then take the argmax of the resulting logits to obtain the corresponding digit.

$i$ cannot receive information from token $j$ through the SSM. Our blocking mechanism is implemented by zeroing out the specific entries in $S$ corresponding to the connections we wish to block. As shown in the Fig. 6, for various anchor pairs, the output after cutting these connections remains nearly identical to the original output, indicating that the SSM plays little role in transmitting this information.

**Information substitution.** To further verify that Mamba relies on convolution for information transmission, we conducted an information substitution experiment: if Mamba encodes all necessary information through convolution, then transferring the resulting state to another sequence should enable it to produce the same output as the original. For all sequences with anchor pairs other

than '43', as illustrated in Fig. 5, we replaced the post-convolution hidden states of the downstream tokens with those from a "43" sequence, where all other elements are identical except for the anchor pair. As shown in Fig. 6, we found that the outputs of nearly all anchor pairs collapse to the output corresponding to the "43" anchor pair.

Taken together, these two experiments clearly demonstrate that Mamba solves the composite function task primarily by leveraging convolution to extract the necessary information. This lays the groundwork for Mamba's inability to reach symmetric solution.

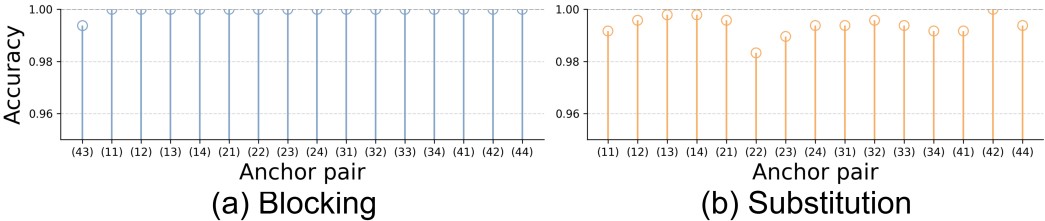

Figure 6: Accuracy of each anchor pair under blocking and substitution experiments compared to the original output. Figures (a) and (b) correspond to the blocking and substitution experiments, respectively. The x-axis denotes the anchor pairs, and the y-axis represents accuracy. Each anchor pair is evaluated using 480 randomly generated sequences.

## 4.2 Nonlinear convolution introduces asymmetry

In this section we will examine that the asymmetry weight in convolution introduces asymmetry between symmetric anchor pairs.

We design a task with fully symmetric setting as follows. The symmetric anchor pairs have the same function but they do not composite by elementary functions; only function "43" is unseen during the training. This leaves the symmetric solution as the only possible one. As shown in the Fig. 7a, we found that standard Mamba can achieve $100\%$ in the test data of "34" function and fails to learn "43" via symmetric property. Additional experimental results are provided in the appendix.

To better isolate the effect of convolution's asymmetry, we remove this asymmetry by setting all convolutional kernel weights to 1. As shown in the Fig. 7b, once the asymmetry of convolution is eliminated, Mamba biases learning the symmetric solution to fit the composite task.

This indicates that Mamba's preference for asymmetry stems from the inherent asymmetry of its nonlinear convolution. In Mamba, sequence information is fused through a nonlinear convolution operation, which serves as the input to the SSM module. Consider two sequences, $(v_1, v_2, v_3, v_4)$ and its reversed counterpart . For the convolution outputs of the above sequences, we define the final token of the result as follows:

original sequence: $f = c_1 \circ v_1 + c_2 \circ v_2 + c_3 \circ v_3 + c_4 \circ v_4 + \beta,$

symmetric sequence: $g = c_1 \circ v_4 + c_2 \circ v_3 + c_3 \circ v_2 + c_4 \circ v_1 + \beta.$

If the convolution parameters $c_i$ are not identical($\beta$ is the bias), then $f$ and $g$ will generally differ. This means that even for token sequences that are symmetric in content, their representations after convolution in Mamba can be significantly different. Such a discrepancy illustrates Mambas inherent asymmetry, as it fails to preserve the equivalence of symmetric inputs.

The root cause of this asymmetry lies in the non-uniformity of the convolution weights. Since the convolution parameters in Mamba are initialized randomly and trained independently, the $c_i$ values typically remain distinct throughout training. As a result, the convolution operation induces a persistent asymmetry, where different token orders lead to different outputs. We examined the cosine similarity between the individual parameters of the convolution kernel at both the beginning and end of training, and found that they were largely orthogonal to one another, indicating a strong and persistent inconsistency throughout the training process. The detailed results can be found in the appendix.

For a Mamba network with initialization rate $\gamma = 0.5$, it cannot learn composite function task by either composite function or symmetric function. We found that if positional encoding is explicitly

included, as shown in the Fig. 7c, such Mamba network learns composite task by symmetric function. Therefore, positional encoding is also critical for learning symmetric solution.

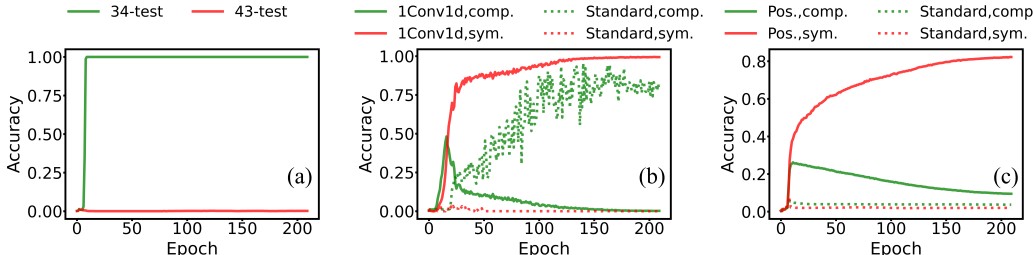

Figure 7: Test Accuracy across. (a): Fully symmetric setting, i.e., target functions are symmetric but do not composite by elementary functions, standard Mamba with $\gamma = 0.5$. (b): Accuracy computed based on groundtruth of composite solution (comp.) and symmetric solution (sym.) for Mamba with all-one convolution (1Conv1d) and standard Mamba (Standard structure) with initialization rate $\gamma = 1$. (c): Similar legend as (b) for Mamba with positional encoding (Pos.) and the standard one but with initialization rate $\gamma = 0.5$. Details of the data and training setup can be found in the appendix.

### 4.3 Transformer with convolution biases asymmetric solution

The empirical analysis in Mamba reveals that the convolution structure is a key component to introduce asymmetry. Imagine if convolution is introduced into the Transformer, does it increase anchor asymmetry and push the model toward learning asymmetric composite solution? We insert a convolution after the input to the Transformer and before the attention module (applying it to $Q$, $K$, and $V$). This is analogous to how Mamba applies nonlinear convolution before the SSM module. For detailed configurations, please refer to the appendix. Previous study(Zhang et al., 2024b) has shown that with different $\gamma$, Transformer can learn symmetric solution or composite solution in different regimes. As shown in Fig. 8, with convolution component, Transformer either can not generalize for small $\gamma$ or fit data by symmetric solution for relative large $\gamma$. In this case, the Transformer exhibits a similar preference for composite function solutions as Mamba and struggles to learn symmetric solutions.

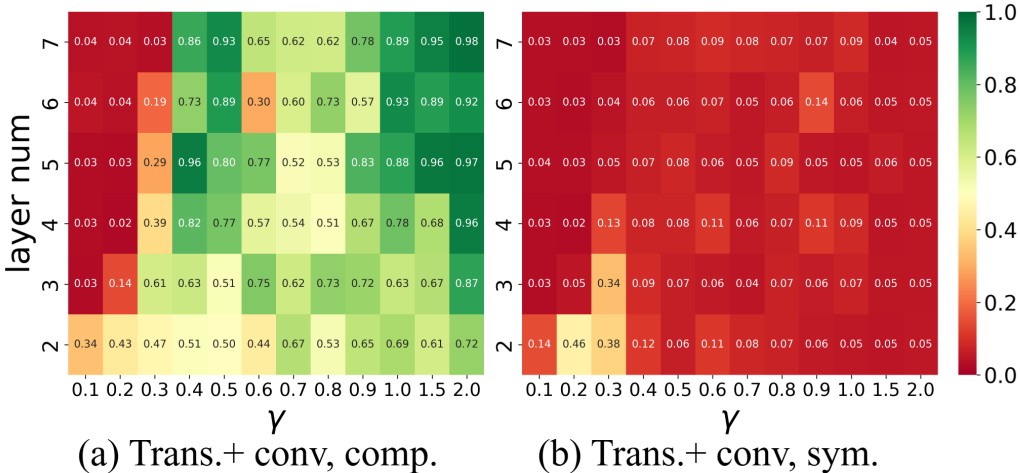

Figure 8: Phase Diagram of the Composite Function Task under Different Settings. Accuracy (color) for composite function task under different initialization rates (abscissa) and depths (ordinate). (a) and (b) show the composite and symmetric solution accuracy of the Transformer after adding nonlinear convolution.

# 5 Mamba's challenges in the inverse sequence matching task

This raises a critical question: Does the change in token order between sequences like 1234 and 4321 make it inherently difficult for Mamba to attend across reversed patterns and discover the correct reverse sequence?

As shown in Fig. 9a, for the standard Mamba network, the training accuracy can easily achieve 100%, however, the test accuracy for the case with tokens seen during training or the case with tokens unseen during the training ("OOD"), the network predicts the outcome with a random-guess level. Additional experimental results can be found in the appendix. For Transformer, or a Mamba network that adds a residual connection from the input of convolution to SSM and the position embedding, as shown in Fig. 9b and 9c, the network can accurately predicts test cases. In addition, for the "OOD" case, the accuracies of such two cases are also significantly larger than random guess. We also conducted additional architectural experiments aimed at mitigating Mamba's asymmetry bias. For details, please refer to the appendix.

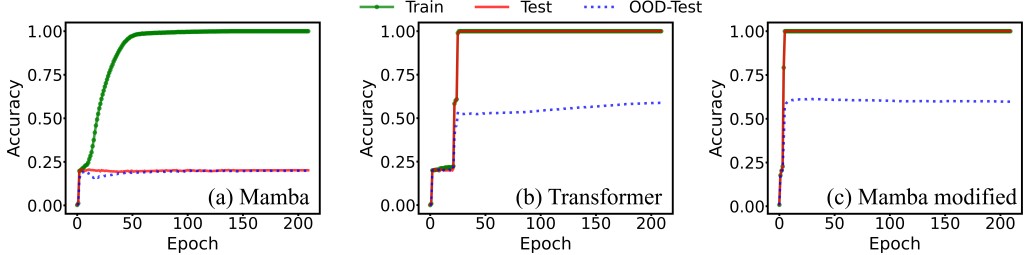

Figure 9: Accuracy on the inverse sequence matching task across different model architectures. (a), (b), and (c) show the accuracy of Mamba, Transformer, and the modified Mamba with residual connection, respectively, on the inverse sequence matching task under the same settings. "OOD" refers to a set drawn from a distribution outside the training and standard test ranges.

# 6 Conclusion

In this work, we leveraged synthetic data to experimentally and systematically analyze the intrinsic properties of the Mamba architecture. Our findings reveal fundamental differences between Mamba and Transformer models, particularly in handling symmetrical patterns and relationships. We identified that Mamba's inherent asymmetry bias stems from its nonlinear convolution mechanism, which fuses token information asymmetrically before passing it to the SSM module. This architectural constraint limits Mamba's ability to recognize symmetrical solutions in composite function tasks and to match reversed sequences. These insights provide valuable guidance for designing future sequence models that combine the computational efficiency of SSMs with the flexible pattern recognition capabilities of Transformers.

# 7 Limitations

To enable a precise investigation and clear illustration of Mamba's internal mechanisms and inductive biases, this work employs synthetic data rather than real-world datasets. While the synthetic tasks are designed to capture key characteristics of the Mamba architecture, they may not generalize to the full diversity of real-world data. Moreover, due to the use of synthetic data, the Mamba models used in our experiments are relatively small. Whether the observed biases persist in larger-scale Mamba architectures remains an open question requiring further investigation. Additionally, many existing large-scale models incorporating Mamba do so in conjunction with Transformer components. In such hybrid architectures, it remains to be studied whether Mamba's inductive bias still dominates or is diminished.

## Acknowledgements

This work is supported by the National Key R&D Program of China Grant No. 2022YFA1008200, the National Natural Science Foundation of China Grant No. 92270001, 12371511, 12422119, 2025 Key Technology R&D Program "New Generation Information Technology" Project of Shanghai Municipal Science and Technology Commission, and the HPC of School of Mathematical Sciences and the Student Innovation Center, and the Siyuan-1 cluster supported by the Center for High Performance Computing at Shanghai Jiao Tong University, and Key Laboratory of Marine Intelligent Equipment and System (Ministry of Education, P.R. China), and SJTU Kunpeng & Ascend Center of Excellence.

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

## A   Experiments compute resources

All experiments were conducted on a server running Ubuntu 22.04.4 LTS. The system is equipped with an Intel Xeon Gold 6133 processor featuring 80 logical cores at 3.0GHz, and 352 GB of RAM. The machine is configured with 8 NVIDIA GeForce RTX 4080 GPUs with 16GB of video memory each.

## B   Detail of Mamba

The following provides a detailed explanation of the internal computations within the Mamba module, beginning with the specification of the corresponding dimensional representations.

$$
\begin{aligned}
B &:= \text{Batch size}, \\
S &:= \text{Sequence length}, \\
D_m &:= \text{dimension of model}, \\
D_{in} &:= \text{dimension of inner model} \\
&= expand \times D_m, \\
N &:= \text{dimension of hidden state}, \\
Num_h &:= \text{number of heads}, \\
H_d &:= \text{dimension of heads},
\end{aligned}
$$

The computations within the Mamba module can be divided into three components: pre-SSM, SSM, and post-SSM. The following presents a detailed derivation for each submodule.

$$\text{Input: } u \in \mathbb{R}^{(\text{B, S, D}_\text{m})},$$

**Pre-SSM**

$$
\begin{aligned}
zxBCdt &= \textbf{Linear}_{\textbf{proj}}(u) \quad \in \mathbb{R}^{(\text{B, S, 2D}_\text{in}+2\text{N}+\text{Num}_\text{h})}, \\
(z, xBC, dt) &= zxBCdt \quad \in \mathbb{R}^{(\text{B, S,}(\text{ D}_\text{in}, \text{ D}_\text{in}+2\text{N, Num}_\text{h}))}, \\
x_c B_c C_c &= \textbf{Conv1d}(xBC) \quad \in \mathbb{R}^{(\text{B, S, D}_\text{in}+2\text{N})}, \\
x_{cf} B_{cf} C_{cf} &= \textbf{SiLU}(xBC) \quad \in \mathbb{R}^{(\text{B, S, D}_\text{in}+2\text{N})}, \\
(x_{cf}, B_{cf}, C_{cf}) &= x_{cf} B_{cf} C_{cf} \quad \in \mathbb{R}^{(\text{B, S,}(\text{ D}_\text{in}, \text{ N, N}))}, \\
A_{init} &\sim \textbf{Uniform}(A_{\min}, A_{\max}) \quad \in \mathbb{R}^{\text{Num}_\text{h}}, \\
A_{\log} &= \log(A_{init}) \quad \in \mathbb{R}^{\text{Num}_\text{h}}, \\
A &= -\exp(A_{\log}) \quad \in \mathbb{R}^{\text{Num}_\text{h}}, \\
dtb_{init} &= \exp\left(\textbf{rand}(\text{Num}_\text{h}) \cdot (\log(\text{dt\_max}) - \log(\text{dt\_min})) + \log(\text{dt\_min})\right) \quad \in \mathbb{R}^{\text{Num}_\text{h}}, \\
dt_{bias} &= dtb_{init} + \log\left(-\exp(-dtb_{init}) + 1\right) \quad \in \mathbb{R}^{\text{Num}_\text{h}}, \\
dt_{fb} &= \textbf{softplus}(dt + dt_{bias}) \quad \in \mathbb{R}^{(\text{B, S, Num}_\text{h})}, \\
\widetilde{A} &= A \circ dt_{fb} \quad \in \mathbb{R}^{(\text{B, S, Num}_\text{h})}, \\
\widetilde{x} &= x_{cf}.\textbf{reshape}(B, S, Num_h, H_d) \circ dt_{fb}.\textbf{reshape}(B, S, Num_h, 1) \quad \in \mathbb{R}^{(\text{B, S, Num}_\text{h}, \text{ H}_\text{d})}, \\
\widetilde{B} &= B_{cf}.\textbf{reshape}(B, S, 1, N) \quad \in \mathbb{R}^{(\text{B, S, 1, N})}, \\
\widetilde{C} &= C_{cf}.\textbf{reshape}(B, S, 1, N) \quad \in \mathbb{R}^{(\text{B, S, 1, N})}.
\end{aligned}
$$

Here, $\textbf{Linear}_{\textbf{proj}}$ denotes a linear transformation, $\textbf{Conv1d}$ refers to a one-dimensional convolution, and $\textbf{SiLU}$ represents the SiLU activation function. $\textbf{Uniform}$ indicates sampling from a uniform distribution, while $\textbf{softplus}$ denotes the Softplus activation function. $\textbf{rand}$ refers to drawing a specified number of random values. The $\circ$ symbol denotes element-wise multiplication, and $\textbf{reshape}$ indicates a dimensional transformation operation.

**SSM**

$$\widehat{A} = \mathbf{Mask}_1 \circ \mathbf{repeat}(\widetilde{A}) \quad \in \mathbb{R}^{(\text{B, Num}_\text{h}, \text{S, S})},$$

$$A_{cumsum} = \mathbf{cumsum}(\widehat{A}) \quad \in \mathbb{R}^{(\text{B, Num}_\text{h}, \text{S, S})},$$

$$L = \exp(\mathbf{Mask}_2 \circ A_{cumsum}) \quad \in \mathbb{R}^{(\text{B, Num}_\text{h}, \text{S, S})},$$

$$P = \mathbf{einsum}("BSHN, BSHN \rightarrow BHSS", \widetilde{C}, \widetilde{B}) \quad \in \mathbb{R}^{(\text{B, 1, S, S})},$$

$$M = \mathbf{einsum}("BHSS, BHSS \rightarrow BHSS", L, P) \quad \in \mathbb{R}^{(\text{B, Num}_\text{h}, \text{S, S})},$$

$$y = \mathbf{einsum}("BHSS, BSHP \rightarrow BSHP", M, \widetilde{x}) \quad \in \mathbb{R}^{(\text{B, S, Num}_\text{h}, \text{H}_\text{d})}.$$

The $\mathbf{Mask}_1$ operation is used to zero out the diagonal and upper-triangular elements of a matrix, retaining only the elements below the diagonal. The $\mathbf{Mask}_2$ sets the elements above the diagonal to negative infinity. The $\mathbf{cumsum}$ operation performs a cumulative sum along the first of the last two dimensions (i.e., across rows) in a matrix of shape $(S, S)$, summing from top to bottom. The $\mathbf{einsum}$ operation denotes the Einstein summation convention, used for concise and flexible tensor contractions.

**Post-SSM**

$$D = \mathbf{1}_{\text{D}_\text{in}} \quad \in \mathbb{R}^{\text{D}_\text{in}},$$

$$\dot{y} = y + x_{cf}.\mathbf{reshape}(B, S, Num_h, H_d) \times D.\mathbf{reshape}(D_{in}, 1) \quad \in \mathbb{R}^{(\text{B, S, Num}_\text{h}, \text{H}_\text{d})},$$

$$\ddot{y} = \dot{y}.\mathbf{reshape}(B, S, D_{in}) \quad \in \mathbb{R}^{(\text{B, S, D}_\text{in})},$$

$$y_z = \ddot{y} \cdot \mathbf{SiLU}(z) \quad \in \mathbb{R}^{(\text{B, S, D}_\text{in})},$$

$$y_{norm} = y_z \cdot \frac{1}{\sqrt{\mathrm{mean}(y_z^2, \mathrm{axis} = -1) + \epsilon}} \cdot \mathbf{w} \quad \in \mathbb{R}^{(\text{B, S, D}_\text{in})},$$

$$y_{out} = \mathbf{Linear}_{\mathbf{proj}}(y_{norm}) \quad \in \mathbb{R}^{(\text{B, S, D}_\text{m})}.$$

Here, $\mathbf{1}$ denotes a vector in which all elements are equal to 1.

$$\text{Output: } y_{out} \in \mathbb{R}^{(\text{B, S, D}_\text{m})}.$$

## C  Data setup

### C.1  Composite function task

**Standard** The total dataset comprises 300,000 samples, and each sequence has a fixed length of 8. Each anchor pair in the training set accounts for $5.6\%$ of the total data, while each anchor pair in the test set constitutes $0.6\%$. The mapping between anchors and their associated functions is as follows: anchor 1 corresponds to a shift of $+5$, anchor 2 to $+1$, anchor 3 to $-2$, and anchor 4 to $-8$. During training, all 15 possible anchor pairs are included except for pair 43. Among these, all pairs except 34 are derived from the composition of their corresponding single-anchor functions. Notably, the function for pair 34 is manually set to $-6$, deviating from the correct compositional result of $-10$. The test set contains both symmetric and compositional instances of pair 43. The distinction between training and test data is governed by the position of the key token. For sequences of length 8, each position is associated with a congruence class modulo 8. In the training set, the key token is prohibited from appearing in the position whose modulo-8 class matches its own. For example, key 33 cannot appear in the first position, as it belongs to the congruence class 1 mod 8, which is aligned with the first index. Conversely, in the test set, each key token is required to occupy the position that corresponds exactly to its modulo-8 congruence class. This design ensures that the model has access to the full semantic range of tokens during training while preventing it from relying solely on positional memorization to generalize to the test set.

**Full symmetry** The total number of datasets is 300,000, and each sequence has a fixed length of 8. Each anchor pair in the training set accounts for $4.5\%$ of the total, while each anchor in the test set accounts for $0.5\%$ of the total. To ensure that the only possible solutions are symmetric,

we simultaneously utilize two sets of correspondences between anchors and functions. This setup prevents the model from simply solving for individual anchor functions and instead forces it to derive symmetric solutions by understanding the symmetry of anchor pairs. There are a total of five anchors: 0, 1, 2, 3, and 4. The 0 anchor is added to balance the data volume between the two function sets, thus avoiding model bias caused by disparities in data quantity. The first set of correspondences between anchors and functions is as follows: 0 corresponds to $+2$, 1 corresponds to $+5$, 2 corresponds to $+1$, 3 corresponds to $-2$, and 4 corresponds to $-8$. The second set of correspondences is: 0 corresponds to $-9$, 1 corresponds to $+6$, 2 corresponds to $-7$, and 3 corresponds to $+3$. All anchor pair functions are composed of either the first or the second set of correspondences. To facilitate the observation of symmetric solutions, anchor pairs 00, 11, 22, 33, and 44 are excluded from this task. Thus, there are a total of 20 anchor pairs in this task. The anchor pairs following the first set of correspondences are 01, 02, 10, 20, 14, 41, 23, 32, 34, 43, a total of 10 pairs. Among them, there are 4 zeros, 4 ones, 4 twos, 4 threes and 4 fours. This design ensures that each anchor has the same amount of data under the two corresponding methods, and the two corresponding methods also have the same amount of data. The anchor pairs following the second set of correspondences are 03, 30, 04, 40, 12, 21, 13, 31, 24, 42. Only 43 is not included in the training set. The purpose of this task is to observe whether the model can discover the symmetry among all anchor pairs and thus output symmetric solutions. The distinction between the training set and the test set in the dataset is based on the position of the key. For sequences of length 8, each position corresponds to a congruence class modulo 8. In the training set, for each key, the key does not fall within the congruence class corresponding to its position. For example, the key 33 cannot appear in the first position of the sequence because it would then be in the congruence class modulo 8 of 1, which corresponds to that position. In contrast, in the test set, each key must exactly fall within the congruence class corresponding to its position. This setup allows the model to learn the meaning of all tokens while preventing it from relying solely on memorization to generalize to the test set.

## C.2 Inverse sequence matching task

The total dataset consists of 100,000 samples. The sequence length varies with the number of layers in the Mamba model. Each sequence is generated from a generation set consisting of three distinct elements. It contains five different permutations of the elements in the generation set, where each permutation is followed by a random token that does not belong to the generation set. Each such permutation is referred to as a key sequence.

One of these five sequences is randomly selected and reversed to form the query sequence, which is appended to the end of the original sequence. Between the key sequences and the query sequence, a number of randomly generated tokensequal to the pure convolutional receptive field of Mamba are inserted to prevent the model from retrieving information through convolution. Therefore, the total sequence length can be formally expressed as follows:

$$S = 5 \times (3 + 1) + 3 \times \mathbf{Num}_{layer} + 3.$$

Here, $\mathbf{Num}_{layer}$ denotes the number of layers in the Mamba model. For example, in the case of a two-layer Mamba model, the sequence length is 29. To ensure a fair comparison, Mamba and Transformer models with the same number of layers are trained on identical sequence configurations.

The training set constitutes 80% of the total dataset, while the test set and the out-of-distribution (OOD) set each account for 10%. In the training set, the three elements in the generating set do not all belong to the same congruence class modulo 3. In contrast, in the test set, all three elements in the generating set belong to the same congruence class modulo 3. For both the training and test sets, all numerical values in the sequences are drawn from the range 20 to 100. In the OOD set, however, all sequence elements are drawn from the range 101 to 200, and the generation set is not subject to any congruence constraints.

# D Training setup

Unless otherwise specified, all tasks in this work adopt the following training parameter settings. The learning rate is initially set to 1e-5 at the start of training, warmed up to 25 times its initial value within 10 epochs, and then decreases to 1e-5 via cosine decay at 200 epochs. The training optimizer is AdamW with parameters set as $\beta_1 = 0.9$, $\beta_2 = 0.999$, $eps = 1e-8$, and weight decay=1e-2.

Meanwhile, gradient clipping is applied with a maximum norm of 1. For composite function tasks, the batch size is set to 2048, while for inverse sequence matching tasks, the batch size is 1024. The loss function used for training is the cross - entropy loss function, which is calculated only for the last token of the model output sequence.

# E   Experiment detail and result

## E.1   Composite function task

### E.1.1   Phase diagram of Mamba on the composite function task

**Mamba configurations**   The dimension of model is set to 32, the dimension of hidden state is set to the default value of 128, and the expansion factor is set to the default value of 2, and the activation function employed throughout the model is the Sigmoid Linear Unit (SiLU). To facilitate clear observation, all experiments are conducted using a single head. The convolution kernel length is set to its default value of 4. The values reported at different positions represent the mean results obtained using three fixed random seeds. Across these positions, the only variations are in the model's depth or initialization.

### E.1.2   Experiments with all-one convolution

**Mamba configurations**   To investigate whether the asymmetry of convolution is responsible for Mamba's difficulty in learning the symmetric solution, we select a model configuration under which Mamba successfully learns the compositional solution but fails to learn the symmetric one. Specifically, we use a five-layer Mamba model with small initialization($\gamma = 1$), while all other settings are consistent with those used in the phase diagram experiments.

### E.1.3   Experiments with positional encoding

**Mamba configurations**   To examine whether Mamba's bias for asymmetry in the composite function task arises from its lack of explicit positional encoding, thus encouraging reliance on convolution. We select a configuration under which Mamba struggles to learn both the composite and symmetric solutions. Specifically, we use a two-layer Mamba model with standard initialization($\gamma = 0.5$). All other settings are kept consistent with those used in the phase diagram experiments.

### E.1.4   Experiment under full symmetry

**Mamba configurations**   To validate Mamba's difficulty in solving the composite function task under fully symmetric settings, we set dimension of model to 128, while keeping all other settings consistent with those used in the phase diagram experiments. The number of layers is varied across 2, 3, 4, 5, and 6, and standard initialization($\gamma = 0.5$) is applied.

**Extended results**   For each configuration, experiments are conducted using three fixed random seeds. The results are shown in the Fig. 10 and Fig. 11. As can be observed, under all configurations, Mamba consistently struggles to solve the composite function task in the fully symmetric setting.

### E.1.5   Cosine similarity among convolution weights

Mamba's asymmetry bias originates from its nonlinear convolution, specifically from the asymmetry of its convolutional kernel parameters. To investigate this asymmetry, we examined the cosine similarity between the convolutional kernels at the beginning and the end of training, as shown in the Fig. 12, and found that they are nearly orthogonal. This indicates that the convolutional parameters exhibit strong asymmetry throughout the training process.

## E.2   Inverse sequence matching task

**Mamba configurations**   The dimension of model is set to 128, the dimension of hidden state is set to the default value of 128, and the expansion factor is set to the default value of 2. The activation

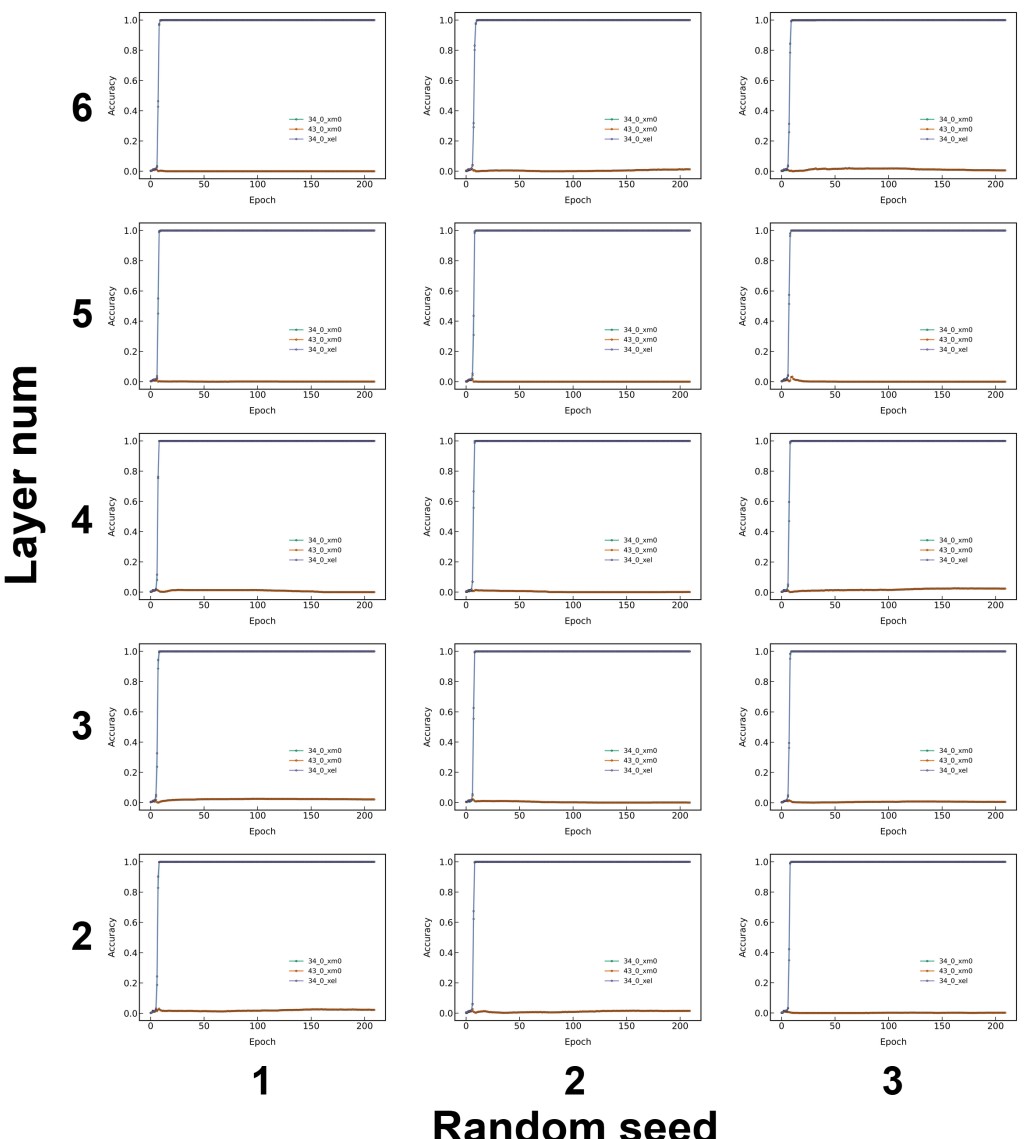

Figure 10: Accuracy of Mamba under fully symmetric setting. The horizontal axis represents the random seed, while the vertical axis corresponds to the number of layers in the Mamba model.

function used is SiLU. To clearly isolate the structural differences between Transformer and Mamba and ensure a fair comparison, the Transformer model also adopts SiLU as its activation function. The convolution kernel length is set to its default value of 4, and the number of heads is fixed at 1.

**Transformer configurations** The dimension of model is set to 128, consistent with the Mamba configuration. To ensure a fair comparison, the Transformer also adopts a $2\times$ dimensional expansion when generating value vectors, mirroring Mambas setup. Specifically, the dimensions of the query and key vectors are set to 128, while the value vectors have a dimension of 256. Additionally, to ensure a fair comparison and maintain a comparable number of parameters between the two models, the Transformers feedforward network (FNN) uses a hidden dimension of 128, and, like Mamba, it is configured with only a single attention head throughout.

**Extended results** All experiments are conducted using the same set of random seeds to ensure fairness in comparison.

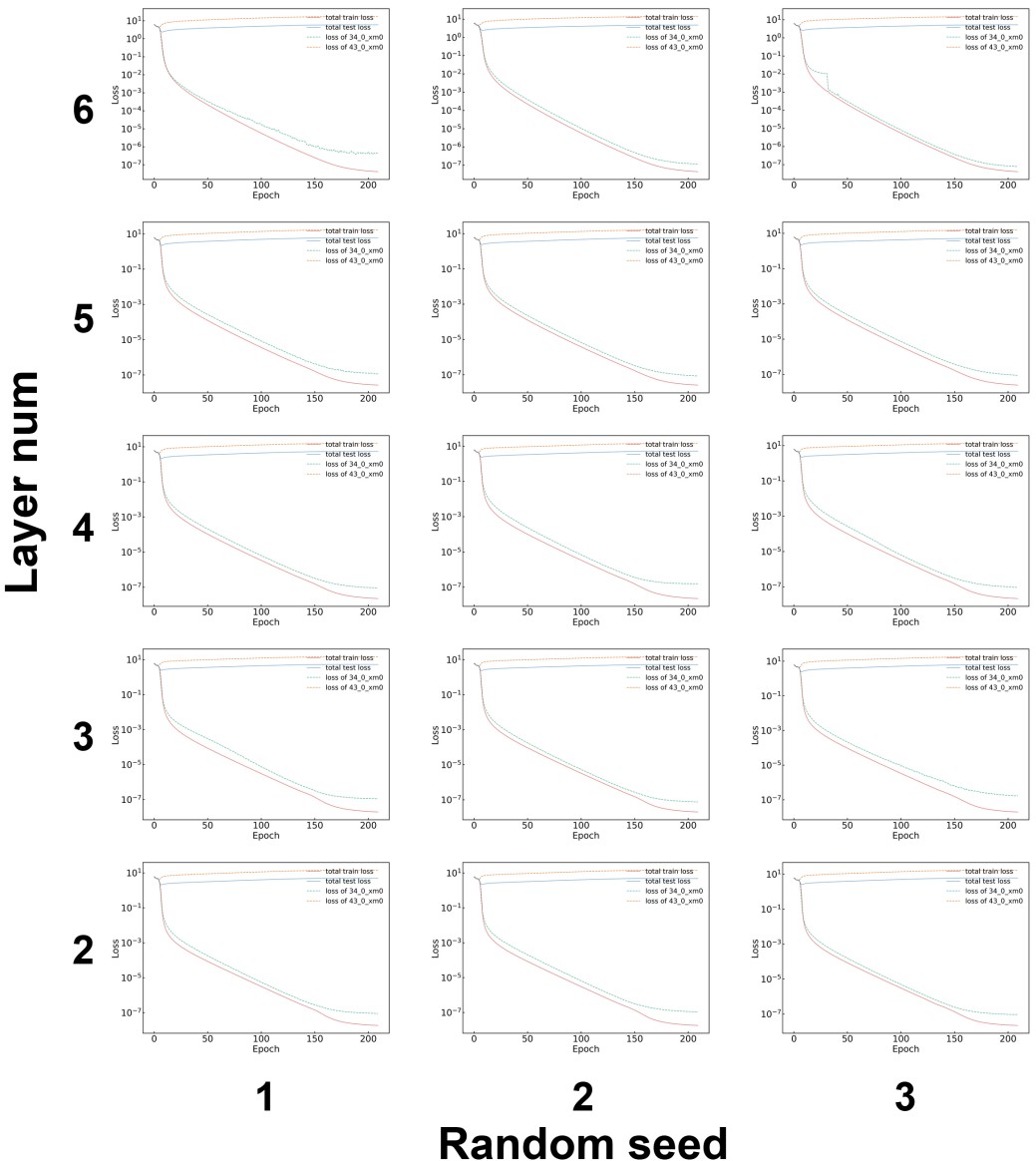

Figure 11: Loss of Mamba under fully symmetric setting. The horizontal axis represents the random seed, while the vertical axis corresponds to the number of layers in the Mamba model.

To evaluate Mambas difficulty in solving the inverse sequence matching task, we vary the model depth across 2, 3, 4, and 5 layers, and consider different initialization strategies ranging from standard to small initialization($\gamma = 0.5, 0.6, 0.7, 0.8, 0.9, 1.0$). The detailed results are shown in the Fig. 13 and Fig. 14. It can be observed that Mamba fails to solve the inverse sequence matching task under nearly all configurations.

For the two-layer Transformer, results under different initialization schemes($\gamma = 0.5, 0.6, 0.7, 0.8, 0.9, 1.0$) are shown in the Fig. 15 and Fig. 16. It can be seen that even with a small number of layers, the Transformer outperforms Mamba.

The results of the modified two-layer Mamba under different initialization schemes($\gamma = 0.5, 0.6, 0.7, 0.8, 0.9, 1.0$) are shown in the Fig. 15 and Fig. 16. It can be observed that the modified Mamba not only significantly outperforms standard Mamba, but also substantially surpasses the performance of the Transformer.

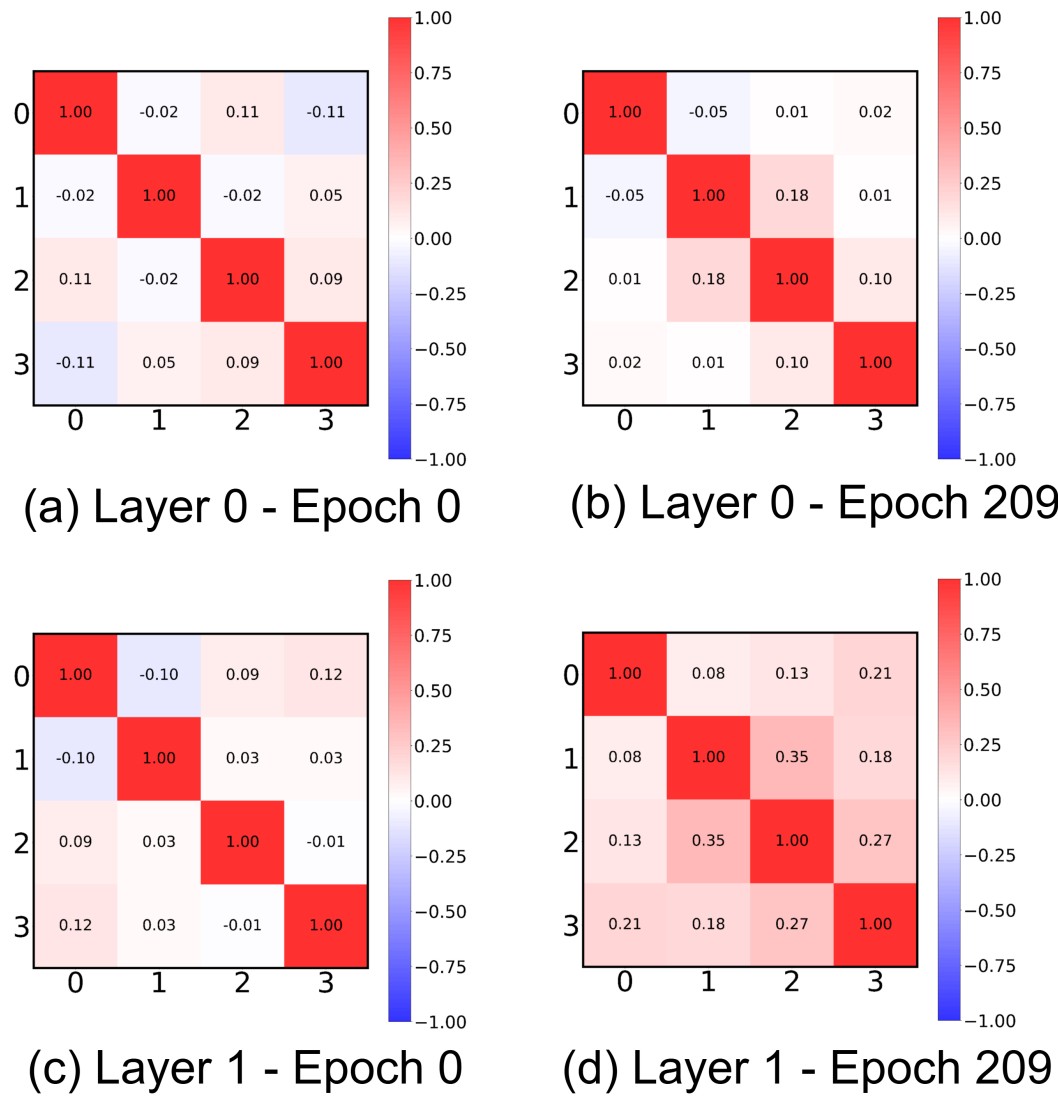

Figure 12: Cosine similarity of convolutional kernel parameters between epoch 0 and epoch 209 for the two-layer Mamba model. The model configuration is consistent with that used in the phase diagram experiments. In this figure, Mamba is configured with two layers and initialized with $\gamma = 1$.

**Experiments with alternative architectural modifications** From our experiments, we observe that Mamba's bias toward asymmetry originates from the inherent asymmetry introduced by its nonlinear convolution. However, by introducing a residual connection that bypasses this nonlinear convolution, the impact of such asymmetry can be effectively mitigated. This allows Mamba to leverage the SSM module for information extraction and even achieve performance comparable to or surpassing that of Transformers on the inverse sequence matching task.

It is important to emphasize that, in order for the SSM to extract information in a manner similar to Attention, positional awareness of tokens is essential. Therefore, positional embedding must be incorporated. In fact, the addition of positional embedding plays a significant role in the inverse sequence matching task as well.

In addition to introducing the residual connection that bypasses the nonlinear convolution, we also conducted another set of architectural experiments. Inspired by the original Mamba architecture, another possible way to inject the raw token information is through a "gating mechanism". For example, one might use a pointwise product between the original token and the fused token (after nonlinear convolution) as the input to the SSM. We additionally conducted multiple experiments on

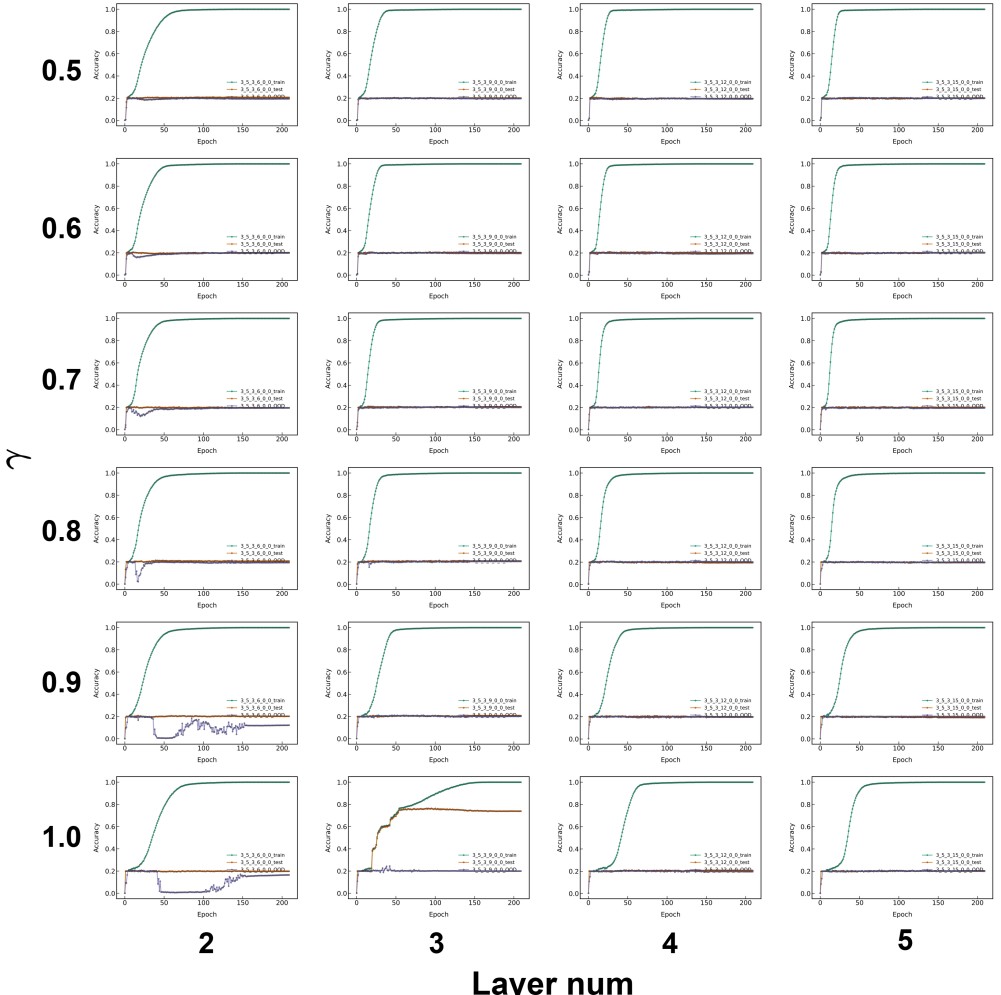

Figure 13: Accuracy of Mamba under different configurations on the inverse sequence matching task. The horizontal axis represents the number of layers in the Mamba model, while the vertical axis corresponds to the initialization scheme.

this approach, and the results are shown in Fig. 17. The following presents the computation process of the gating mechanism. Given the input $U$ to the Mamba block, we have:

$$
\begin{aligned}
(\tilde{U}, Z, dt) &= \text{Linear}(U), \quad U \in R^{(s,d)}, \tilde{U} \in R^{(s,2d+2h)}, Z \in R^{(s,2d)}, dt \in R^{(s,N_h)}, \\
(B, C, X) &= \sigma(\text{Conv1d}(\tilde{U})), \quad B \in R^{(s,h)}, C \in R^{(s,h)}, X \in R^{(s,2d)}, \\
(\tilde{B}, \tilde{C}, \tilde{X}) &= (B, C, X) \circ \tilde{U}, \quad \tilde{B} \in R^{(s,h)}, \tilde{C} \in R^{(s,h)}, \tilde{X} \in R^{(s,2d)}.
\end{aligned}
\tag{10}
$$

Finally, $\tilde{B}$, $\tilde{C}$, and $\tilde{X}$ serve as the input to the SSM.

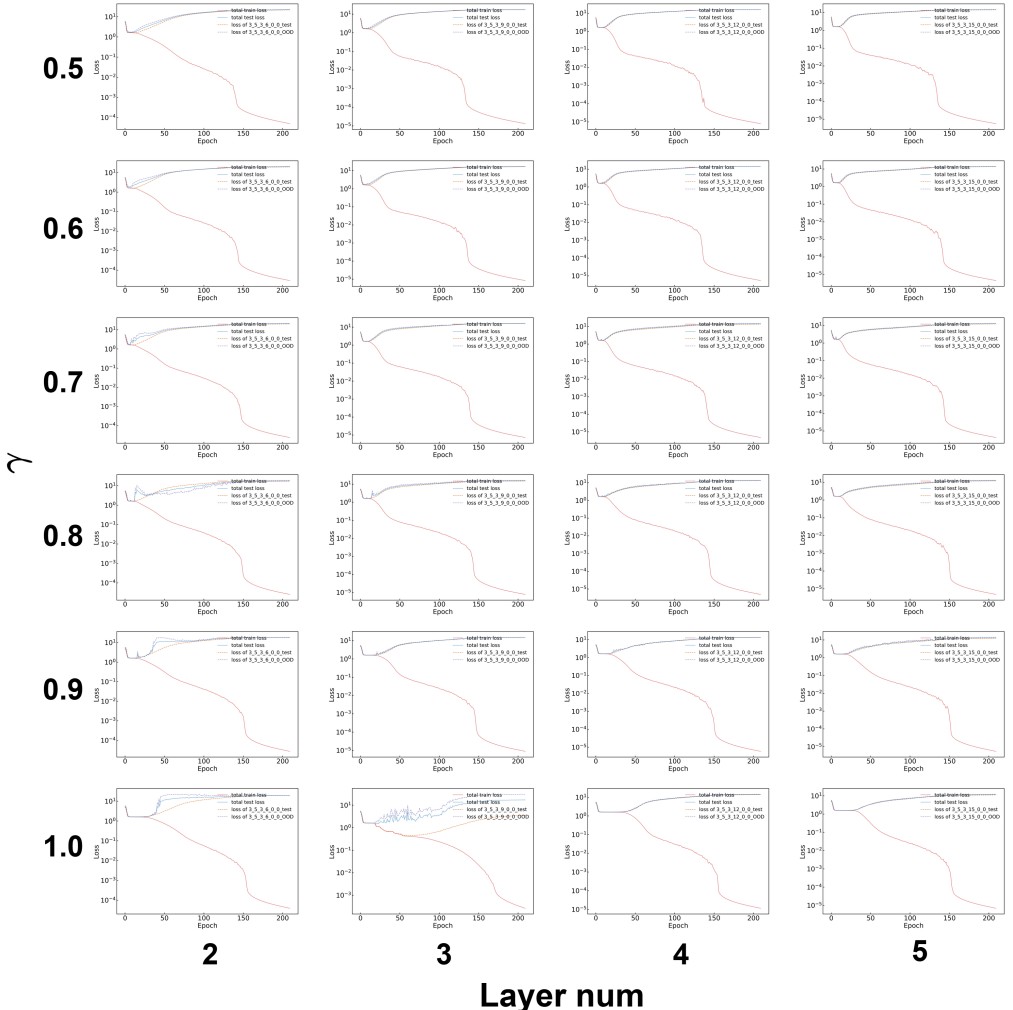

Figure 14: Loss of Mamba under different configurations on the inverse sequence matching task. The horizontal axis represents the number of layers in the Mamba model, while the vertical axis corresponds to the initialization scheme.

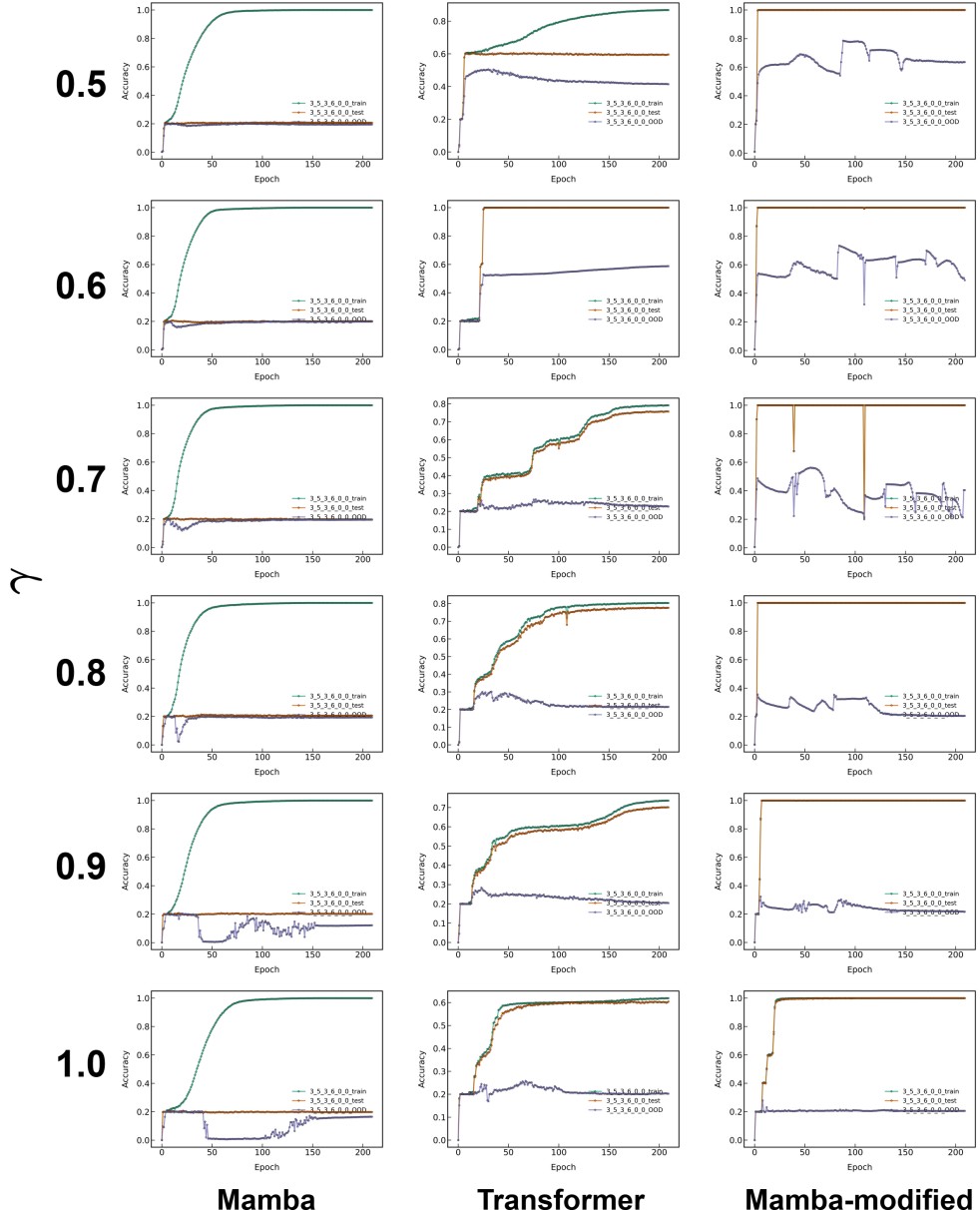

Figure 15: Accuracy of Mamba, Transformer, and modified Mamba on the inverse sequence matching task. Left: Mamba (2-layer), Center: Transformer (2-layer), Right: Modified Mamba (2-layer); all under varying initialization schemes.

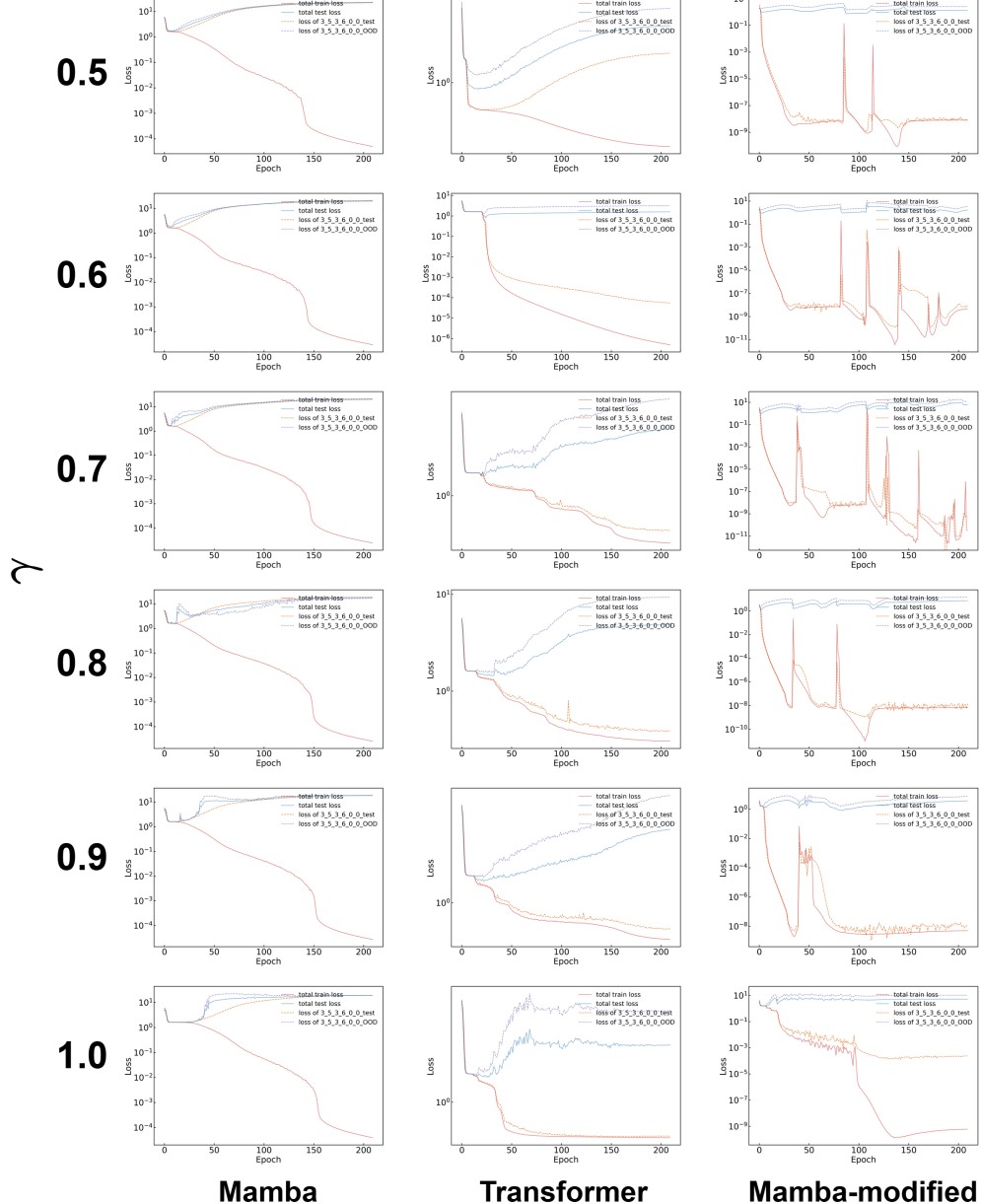

Figure 16: Loss of Mamba, Transformer, and modified Mamba on the inverse sequence matching task. Left: Mamba (2-layer), Center: Transformer (2-layer), Right: Modified Mamba (2-layer); all under varying initialization schemes.

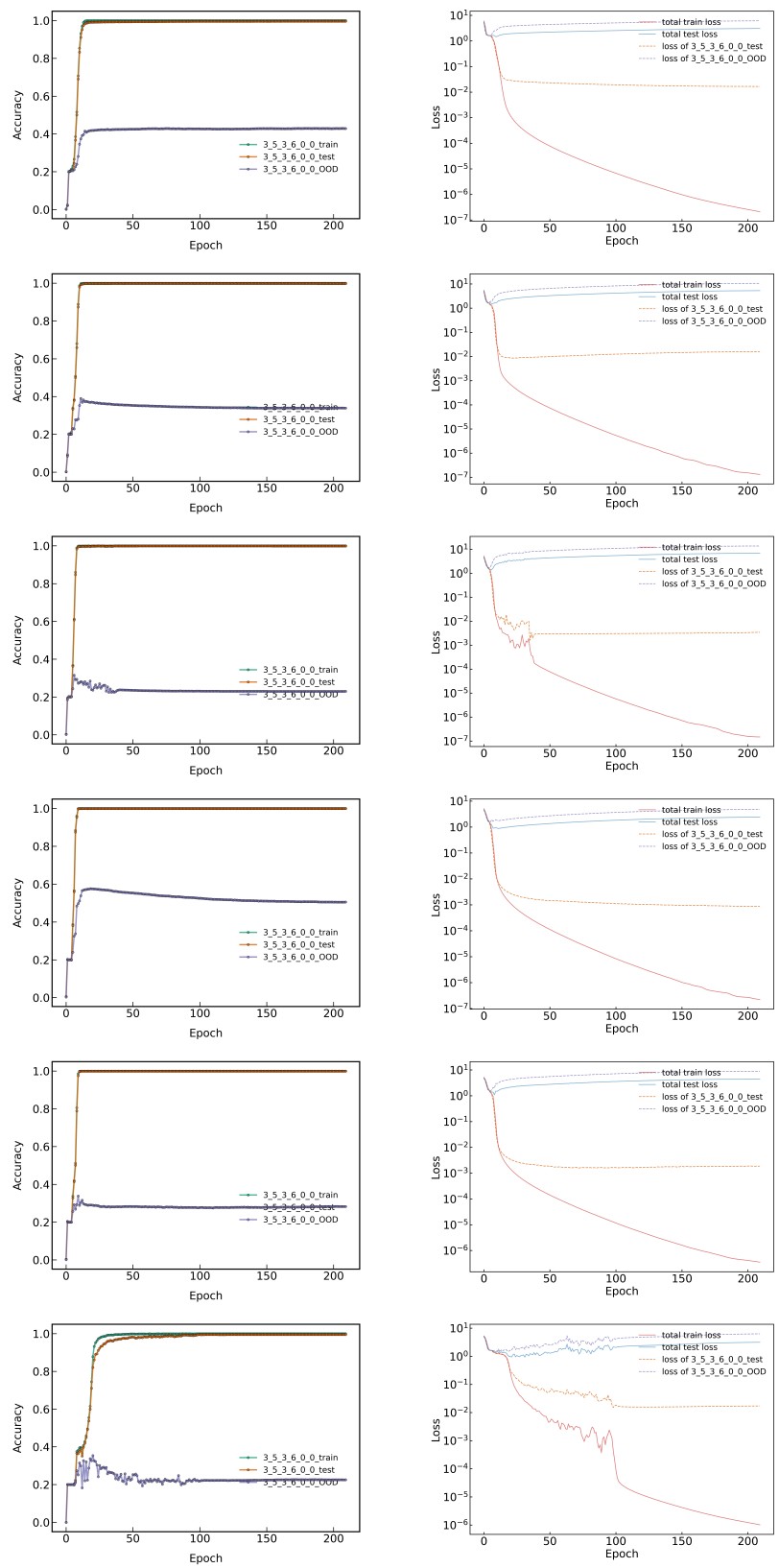

Figure 17: Accuracy (left) and loss (right) curves of Mamba with gate residual connections on the inverse sequence matching task. The architecture of gate-residual Mamba is consistent with that of residual Mamba. From top to bottom, the initialization is set to $\gamma = 0.5$, $\gamma = 0.6$, $\gamma = 0.7$, $\gamma = 0.8$, $\gamma = 0.9$, and $\gamma = 1.0$.

