# OpenReview forum: "Achilles' Heel of Mamba: Essential difficulties of the Mamba architecture demonstrated by synthetic data"
_NeurIPS.cc/2025/Conference — NeurIPS 2025 spotlight_

### Official Review · Reviewer_7EnS · 2025-06-15

**Clarity:** 2
**Significance:** 3
**Originality:** 2
**Rating:** 5
**Confidence:** 4

**Summary:**

This paper presents evidence for the “asymmetric bias” introduced by the convolution in Mamba. In particular, the authors focus on two setups: composite function and inverse sequence matching. They compare Mamba with Transformers and show that Mamba learns qualitatively different solutions due to the convolution prior to the SSM part. To further show this, the authors also modify Mamba by setting all convolution weights to one or adding residual connection, both interventions mitigate such bias.

**Questions:**

Q1 Why is the bias introduced by the convolution bad? The authors only report that it hurts performance in the inverse sequence matching task. Additional evidence or discussion, possibly in a more realistic setup, can strengthen the paper.

Q2 Introducing residual connections seems to greatly reduce the convolutional bias and allows Mamba to learn the inverse sequence matching task. What is its effect in the composite function task? Since this modification is easy, would you recommend it for future Mamba-like models? An experiment showing e.g. that the residual connection does not hurt pretraining could strengthen the paper.

Q3. Can the author clarify the information blocking setup? How is the SSM matrix defined? How is the blocking implemented? I feel this is not clear from the text and Figure 5.

**Comments on the clarity and minor comments:**

C1 Missing word in line 106

C2 Sections 3.1 and 3.2  contain many typos, e.g. the input to the SSM should contain $A$ or $\Delta_t$ and not $H$, also including (at least comments on) the dimensionality of the input could help readers.

C3 The Receptive field of a 2 layer mamba is said to be 6 in line 157. I assume this is because the size of the convolution kernel is 3. However this is not true since the second convolution comes after the SSM part so its receptive field is potentially all the previous tokens.

C4 Figure 5: Layer 1 output is written in natural numbers as the input but shouldn’t it be real vectors in this case?

C5 Line 178: SSM matrix definition should be included or better explained. I don't think the expression in figure 5 (referred to as flow) is correct since the effect of A should also be included.

C6 Figure 6 could be greatly reduced in size and the y axis could e.g. show the interval 0.95-1.

C7 Line 217: “Previous study has shown”, a citation could help here.

C8 Line 218-219 are unclear: the transformers seem to have the same behaviour as mamba when equipped with the convolution, the text should reflect that.

C9 I think there is a typo in Figure 7: standard mamba should be $\gamma=0.5$, not $\gamma = 1$.

**Ethical Concerns:**

["NO or VERY MINOR ethics concerns only"]

**Final Justification:**

The authors addressed my concerns about clarity and provided an experiment with commercial-grade LLMs, albeit not so realistic. Nonetheless, the work has identified an interesting bias of the causal convolution and an effective way to mitigate it, therefore I recommend acceptance.

**Limitations:**

The authors include a limitation section after the conclusion discussing the lack of real-world experiments and experiments at a larger scale.

**Quality:**

3

**Strengths And Weaknesses:**

**Strengths**

S1 Experimental evidence supports the proposed hypothesis: all experiments performed strongly suggest that the convolution indeed biases Mamba’s solution.

S2 Most new architectures based on SMMs include a 1d convolution, while standard Transformers do not. Hence,  this study is important since it sheds light on a possible problem of using the convolution.

**Weaknesses**

W1 The paper is quite difficult to read and contains many typos. Some sections could be expanded to explain things more clearly.

W2 Experimental setup is limited and the paper lacks experiments on natural language on a realistic task to show the hurting effect of the convolution.

---

> ### Author Rebuttal · Authors · 2025-07-29
>
> Through revisions and experiments based on each of your suggestions, our work has been substantially improved. We sincerely thank you for your valuable guidance.
>
> **W1:**
>
> In line with your suggestion, we will revise Section 3.1 as follows: as per your suggestion, we will replace $H$ with $X$ to facilitate comparison. In the previously submitted version, we omitted $t$ and $A$ in the equations to emphasize the input to the SSM, but we will now include them in the updated version for completeness. In addition, we have provided explicit dimension annotations to aid clarity.
>
> **Section3.1:** The Mamba block can be divided into three parts: the widely known SSM (State Space Model) component, the pre-SSM part, and the post-SSM part. Omitting trivial dimension transformations and setting the batch size to 1 to omit the batch dimension, for a given input
> $U$ to the block, the internal computation process to obtain the output
> $O$ can be described as follows:
>
> Pre-SSM
>
> $( \tilde{U}, Z, dt) = \textbf{Linear}(U),~~~ U\in R^{(s,d)}, \tilde{U}\in R^{(s,2d+2h)},Z\in R^{(s,2d)}, dt\in R^{(s,N_h)},$
>
> $(B, C, X) = \sigma(\textbf{Conv1d}(\tilde{U})),~~~ B\in R^{(s,h)}, C\in R^{(s,h)}, X\in R^{(s,2d)},$
>
> SSM
>
> $Mask = \textbf{F}(A_0, dt),~~~ A_0\in R^{N_h}, Mask\in R^{(N_h, s, s)},$
>
> $I = \textbf{Repeat}(C B^{\top},N_h)),~~~  I\in R^{(N_h, s, s)},$
>
> $S = Mask\circ I,~~~  S\in R^{(N_h, s, s)},$
>
> $Y = SX + X,~~~ Y\in R^{(N_h, s, 2d/N_h)},$
>
> Post-SSM
>
> $Y_{Norm} = \textbf{RMS}(Y\circ\sigma(Z)),~~~ Y_{Norm}\in R^{(s, 2d)},$
>
> $O = \textbf{Linear}(Y_{Norm}),~~~ O \in R^{(s, 2d)},$
>
> where
> > - $S$ : SSM matrix,
> > - $s$ : sequence length,
> > - $d$ : hidden‑state dimension,
> > - $h$ : SSM hidden dimension,
> > - $N_h$ : number of SSM heads,
> > - $\textbf{Linear}$ : linear transformation,
> > - $\textbf{Conv1d}$ : one‑dimensional convolution,
> > - $\sigma$ : nonlinear activation function,
> > - $\textbf{F}$ : function generates the $Mask$,
> > - $\textbf{Repeat}$ : dimension‑replication operation,
> > - $\circ$ : pointwise multiplication,
> > - $\textbf{RMS}$ : RMS normalization.
>
> In response to your suggestion, we have revised and expanded Section 3.3, incorporating real-world examples to make the composite function task and its underlying symmetry more intuitive and accessible to readers. Also, we have expanded the Section3.4 and added real-world analogues of the task to help readers better understand the inverse sequence matching task and its fundamental nature. Owing to space constraints, we omit the detailed enumeration here.
>
> **W2:**
>
> We sincerely value your guidance, and to verify the real‑world relevance of our findings, we converted the inverse sequence matching task proposed in the paper into a natural‑language version and evaluated it under the most practical setting. Specifically, we ran the test 3 times on two commercial LLMs: Hunyuan‑TurboS and DeepSeek‑V3. Hunyuan‑TurboS contains 57 stacked Mamba blocks, while DeepSeek‑V3 consists of Transformers and has a comparable parameter count and overall benchmark performance. Each run contained 100 randomly generated inverse sequence matching problems.
>
> Across the three runs, Hunyuan‑TurboS achieved an average accuracy of **17.6\%**, far below DeepSeek‑V3’s average of **52.3\%**. These results mirror the trends we observed on smaller models in the paper, clearly demonstrates the impact of the asymmetry bias introduced by nonlinear convolution on the practical performance of Mamba in large-scale models and further reinforcing our conclusions and their practical significance.
>
> A real testing example can be found in our response to Reviewer  UvAr **Q1**.
>
> **Q1:**
>
> We sincerely appreciate the thoughtful suggestions you provided. Through the composite function task, we revealed Mamba’s intrinsic bias for asymmetry—a fundamental property introduced by its convolutional design. In our view, no property is inherently “bad”; rather, some are simply ill‑suited to particular applications.
>
> This bias becomes a shortcoming when the task demands understanding or handling symmetry—as in the inverse sequence matching task. However, in tasks where symmetry is unnecessary, or even needs to be avoided, Mamba’s bias may turn into an advantage. For instance, “finding Jack’s mother’s phone number” constitutes an asymmetric composite function task, whereas its symmetric inversion—“finding the phone number’s mother of Jack”—is both meaningless and distinct from the original. In such tasks, Mamba’s asymmetric bias ceases to be a drawback; other biases or features will instead govern its differences from the Transformer.
>
> As further evidence, we have validated these observations by implementing the inverse sequence matching task in a commercial scale large language model using natural‑language prompts. The full details are provided in Section W2.
>
> **Q2:**
>
> We are deeply grateful for your question.
>
> **Res needs to work with SSM:** After introducing a residual connection, the SSM can draw on two information sources in its input: (i) the fused, inherently asymmetric representation produced by the convolution, and (ii) the un‑fused, non‑linearly untransformed signal carried through the residual path. The advantage is that, in tasks such as inverse sequence matching task—which require the SSM to match two symmetrical signals—the model can lean on the residual connection, thereby mitigating the asymmetry introduced by the convolution.
>
> **SSM's non-functioning $\rightarrow$ Res do not work on composite function task:** We applied this residual‑augmented Mamba to an extensive set of composite function experiments. The results show that, compared with the standard Mamba, the modified Mamba attains composite solution over a larger region of settings, yet it still almost never converges to symmetric solutions. The reason is that, in the composite function task, Mamba can extract the necessary information directly through the convolution, whereas in the inverse sequence matching task the convolutional path is blocked by intervening tokens, forcing the model to rely solely on the SSM. When Mamba can acquire information directly through its convolution—as our experiments demonstrate—it may rely exclusively on that convolutional path, bypassing the SSM entirely. Consequently, the residual pathway we introduced to support the SSM no longer serves a meaningful function in this scenario. Thus, the model’s asymmetric bias remains unchanged.
>
> We will recommend adding a residual connection when employing Mamba. For longer‑sequence problems like inverse sequence matching task (and many real‑world tasks are far longer), Mamba cannot rely on the convolution alone; the residual then becomes a critical alternative pathway that helps the model gather and align information. We are eager to validate our findings with large scale pre‑training in the future, but in the near term we may not have sufficient computational resources to do so and hope for your understanding.
>
> **Q3 & C5:**
>
> As you kindly suggested, we will revise Section 3.1 and the caption of Figure 5, and we will add an appendix subsection to explain our blocking mechanism in greater depth so that readers can follow the details more easily.
>
> Our blocking strategy is built on Mamba’s SSM matrix, defined as
>
> $S = Mask\circ (CB^{\top})$ ,
>
> and the output of SSM block is
> $SX$.
>
> The matrix $S$ describes information flow within the SSM. After omitting the head dimension, its shape is $(s,s)$, where $s$ is the sequence length. The element $S_{ij}$ denotes the flow of information from token $j$ to token $i$; if $S_{ij} = 0$, no information passes from token j to token i through the SSM. Our blocking method simply zeroes the entries in S corresponding to flows we wish to block.
>
> **C1:**
>
> **line 106:** The design of our composite function task directly follows the approach used in Zhang et al.
>
> **C2:**
>
> We will correct these points in Section 3.1. A revised version—including detailed dimension descriptions—has already been provided in our response to W1.
>
> **C3:**
>
> We sincerely apologize for the confusion our oversight has caused. We mistakenly used “receptive field” when we should have written “pure convolutional receptive field.” Here, the term “convolutional receptive field” refers exclusively to how many tokens can be accessed through convolution alone, without invoking the SSM. We wish to prevent Mamba from obtaining information purely via convolution rather than through the SSM. We will replace all incorrect expressions in the original manuscript with "pure convolutional receptive field."
>
> **C4:**
>
> The reason the output of layer1 appears as a numeral is that we pass the output of laye 0 through the model’s final linear layer and then take the arg‑max of the resulting logits to obtain the corresponding digit. This approach enables us to better interpret the model’s intermediate representations and facilitates visualization. We will briefly explain this procedure in the caption of Figure5 and add a dedicated subsection in the appendix for a more detailed description.
>
> **C6:**
>
> We will reduce the figure size and adjust the y‑axis to the range 0.95–1.
>
> **C7:**
>
> We will add the appropriate reference in the relevant section.
>
> **C8:**
>
> **line 219:**  In this case, the Transformer exhibits a similar preference for composite function solutions as Mamba and struggles to learn symmetric solutions.
>
> **C9:**
>
> We apologize for the imprecise wording in our earlier explanation. Here, we are referring to the standard Mamba architecture initialized to 1. With the default initialization of $\gamma = 0.5$, Mamba is unable to learn either the composite or the symmetric solution, making it impossible to observe any bias. By setting the $\gamma$ to 1, Mamba learns the composite solution much more reliably. Therefore, we employ an initialization of $\gamma = 1$. We will revise the caption of Figure 7 accordingly.

---

> > ### Comment · Reviewer_7EnS · 2025-08-03
> >
> > Thank you for the response.
> >
> > The points about clarity have been mostly addressed by the changes promised by the authors. Although it would be helpful to include the definition of the function $\textbf{F}$ that generates the mask in the final manuscript.
> >
> > The additional real-world experiment demonstrates that the identified bias is present also in very large commercial-grade models and is therefore valuable. However, it is still not a very realistic setup, but a natural language version of one of the proposed synthetic tasks.
> >
> > I am raising the score from 4 to 5.

---

> > > ### Author Response · Authors · 2025-08-04
> > >
> > > Thank you sincerely for recognizing our work and offering such valuable feedback.
> > >
> > > The appendix will now include full calculations and a precise definition of $\textbf{F}$. Your careful guidance clarified many details, and we are grateful.
> > >
> > > Since the rebuttal window was brief and real-world language tasks are complex, we converted our experiments into natural-language form and tested them with large language models. This is only an initial step; we remain committed to evaluating the models under fully authentic conditions.
> > >
> > > Your thoughtful review has been invaluable, and we deeply treasure the fruitful exchange it has fostered.

---

### Official Review · Reviewer_2GPi · 2025-06-29

**Clarity:** 1
**Significance:** 1
**Originality:** 3
**Rating:** 3
**Confidence:** 3

**Summary:**

This paper studies the Mamba architecture by testing it on a designed synthetic tasks.
The authors show that Mamba generalizes to learn composite and not symmetric solutions in that task. They identify the source of this generalization as the nonlinear convolution layer. The paper compares Mamba to Transformers and suggests a simple change (adding a residual connection) that helps Mamba learn the symmetric solution

**Questions:**

Could you clarify the blocking mechanism used in your experiments? The description in the paper is too brief to understand how information was actually blocked. There has been related work (such as Endy et al., "Mamba Knockout for Unraveling Factual Information Flow") that explores more targeted and interpretable blocking methods in Mamba-based models. It would be helpful if you could explain whether your method aligns with or differs from such approaches.

**Ethical Concerns:**

["NO or VERY MINOR ethics concerns only"]

**Final Justification:**

Repeating here what I wrote to the reviewers:

Regarding W1, I am still not entirely satisfied with the authors' response. While I now better understand the problem and acknowledge that the authors demonstrate differences in performance in real-world "large" models, I still struggle to see the relevance of the synthetic task (as presented in the rebuttal to Reviewer UvAr, Section Q1) to real-world language problems. However, I admit that I am not an expert on such synthetic tasks, so I might be setting the bar too high. Given my lingering doubts about the importance of this synthetic task, it is challenging for me to give a high recommendation for acceptance. *Nonetheless, I note here explicitly that since I am not an expert on synthetic tasks, if the Area Chair believes the task is indeed important, they may disregard this particular point.*

Regarding W2, the paper's lack of clarity remains a significant issue. However, I acknowledge and appreciate that the authors have committed to substantially improving clarity in the manuscript.

Considering the anticipated improvements in clarity and recognizing that another reviewer sees the synthetic task as important, I am raising my score to a 3.

**Limitations:**

yes

**Paper Formatting Concerns:**

None.

**Quality:**

3

**Strengths And Weaknesses:**

**Strengths**
* The paper focuses on a specific synthetic problem and carefully investigates how Mamba performs on it.

**Weaknesses**
* I was not convinced that the proposed synthetic tasks are meaningful or impactful for the wider research community. The paper does not make a strong case for why these tasks matter, either in terms of practical relevance or future research directions, moreover they don't explain why learning a composite solution is bad. If this is based on earlier work, the authors should clearly restate why these tasks are important.
* Some parts of the text are unclear. For example, Section 3.1 (Introduction of Mamba) and Section 3.3 are hard to understand on their own. The figures are missing captions, which makes them harder to interpret. Specifically, the framing of the composite function task in Section 3.3 is confusing, and the cited prior work addresses slightly different problems.
* The blocking mechanism is not well explained. It would help if the authors provided more detail on how it was implemented and why it works.

---

> ### Author Rebuttal · Authors · 2025-07-29
>
> We sincerely appreciate your suggestions. Guided by your comments, we have meticulously revised the manuscript, performed additional experiments, and addressed all of your comments in detail.
>
> **W1:**
>
> We thank the you for the insightful feedback. They raised a crucial point about the need to clearly establish the significance of our synthetic tasks and their practical relevance. To directly address this, we have conducted new experiments on large-scale models which demonstrate that the architectural biases revealed by our synthetic tasks have tangible, real-world consequences.
>
> - **Bridging Synthetic Tasks to Real-World Impact:** The primary motivation for our synthetic task was to isolate and understand a fundamental architectural property—symmetry—in a controlled environment. We hypothesized that Mamba's inherent asymmetry would lead to challenges in tasks requiring symmetrical or inverse reasoning.
>
>   To validate this, we tested our hypothesis on two prominent, large-scale language models: Tencent’s Hunyuan-TurboS (a hybrid model with 57 Mamba layers) and DeepSeek-V3 (a Transformer-based model of comparable scale). We designed a challenging inverse sequence matching task that requires the model to find a key and return its corresponding value in reverse order. The results are striking:
>
> | Model          | 1st. round Acc. | 2nd. round Acc. | 3rd. round Acc. | Avg. Acc. | Avg. SER|
> | - | - | -| - | - | - |
> | Hunyuan‑TurboS          | 19\%     | 17\%     |17\%      |17.6\%      |19.2\%|
> | DeepSeek-V3 | **47\%**     | **56\%**     | **54\%**     | **52.3\%**     |**3.8\%**|
> Avg. SER is the symbol error rate on correctly indexed retrievals.
>
>   The significant performance gap between the Mamba-containing model and the Transformer model provides strong evidence that the architectural asymmetry we identified is not a mere theoretical curiosity. It has a clear, negative impact on performance in a practical, retrieval-style task that mimics challenges found in complex reasoning. A real testing example can be found in our response to Reviewer  UvAr **Q1**.
>
> - **The Rationale for Our Synthetic Tasks and Their Implications**
> Having established the practical impact, we wish to clarify the rationale for our methodology:
>   - **Why use synthetic composite functions:** These functions are fundamental constructs in mathematics and computer science. A model's ability to learn them, especially with regard to properties like symmetry, serves as a powerful diagnostic for its underlying reasoning capabilities. As the reviewer suggested, this approach is grounded in prior work (e.g., Zhang et al. [1], Hang et al. [2]), which has successfully used such tasks to probe the emergent abilities of LLMs.
>
>   - **Architectural bias is critical:** The reviewer asked why failing at this is "bad." It's not about a universal "good" or "bad" but about fitness for certain tasks, as the no free lunch theorem demonstrates. Our findings show that Mamba's architectural bias results in a specific, measurable deficit in tasks requiring symmetrical information mapping (e.g., A -> B and B -> A). The poor performance on the inverse sequence matching task is the direct consequence. This limitation is critical for applications that rely on bidirectional associations or logical inversions.
>
> **W2:**
>
> We thank the reviewer for their detailed feedback on the clarity of our manuscript. We acknowledge that Sections 3.1 and 3.3, along with the figures, need improvement. As suggested, we will thoroughly revise these sections and add comprehensive captions to all figures in the updated manuscript to make our methodology much easier to follow. To address the reviewer's specific concern about the composite function task, we offer the following clarification on its framing and purpose.
>
> **composite function task:** Anchors 1, 2, 3, and 4 (depicted in orange) each represent distinct functions. Among the 16 anchor pairs formed, 14 correspond to composite functions derived directly from the sequential application of the individual anchor functions. The pair "34", highlighted in red, is defined as a different function rather than a direct composition. The pair "43" is intentionally excluded from the training set. The input to each anchor pair function is referred to as the "key" (indicated in green).
>
> **A Simple Analogy:**
>
> The core idea of our task can be understood with a simple real-world analogy. Imagine a row of people indexed by numbers. Let's define two functions: $n(x)$:  Find the person n positions to the right of person $x$ and $m(x)$: Find the person m positions to the left of person $x$. The composite function $m(n(x))$ means "start at person x, move n steps right, then move m steps left." A model can solve this in two ways:
> - Compositional Reasoning: Execute the steps sequentially.
> - Symmetry-based Shortcut: Recognize that $m(n(x))$ is equivalent to moving $n−m$ steps right and that its symmetric counterpart, $n(m(x))$, yields the same result.
> Our task is designed to discover which strategy a model prefers. Moreover, its fundamental goal—evaluating compositional generalization—is deeply connected to established benchmarks like SCAN and COGS [10, 11]. These benchmarks also test a model's ability to combine known components into novel structures.
>
> **W3 & Q1:**
>
> We appreciate the reference to [5]. This is an excellent paper that provides a more targeted, surgical method for analyzing information flow specifically within Mamba's architecture.
>
> However, as our manuscript was submitted on May 15, 2025, this work, which was posted on arXiv on May 30, 2025, was published concurrently and was thus not available to us at the time of writing.
>
> Furthermore, our goal required a simple, model-agnostic method to validate long-range reasoning in both the Mamba-based model and the Transformer [12]. The common technique of inserting random tokens was the most direct and appropriate choice for this purpose.
>
> **Our Blocking Mechanism: A Common and Effective Technique**:
>
> The SSM mechanism can be expressed as
>
> $Y=SX+X$, with $S=Mask\circ(CB^{\top})$,
>
> where the matrix $S$ has a shape of $(s,s)$, assuming the head dimension is omitted and $s$ is the sequence length. The $(i,j)$ element of $S$ represents how much token $i$ attends to token $j$. If this value is set to $0$, it implies that token $i$ cannot receive information from token $j$ through the SSM. Our blocking mechanism is implemented by zeroing out the specific entries in $S$ corresponding to the connections we wish to block.
>
> We will update the manuscript to include this detailed explanation and properly situate our method within the context of established practices.
>
> **References**
>
> [1]Zhang et al. Initialization is Critical to Whether Transformers Fit Composite Functions by Reasoning or Memorizing. NeurIPS 2024.
>
> [2]Hang et al. Scalable Complexity Control Facilitates Reasoning Ability of LLMs. arXiv: 2505.23013 (2025).
>
> [3]Xu et al. Overview frequency principle/spectral bias in deep learning. Commun. Appl. Math. Comput. 7 (3): 827–864 (2025).
>
> [4]Dao et al. Transformers are SSMs: Generalized Models and Efficient Algorithms Through Structured State Space Duality. arXiv:2405.21060 (2024).
>
> [5]Endy et al. Mamba Knockout for Unraveling Factual Information Flow. arXiv:2505.24244 (2025).
>
> [6]Wolpert et al. No free lunch theorems for optimization. IEEE Trans. Evol. Comput., 1 (1), 67–82. (1997).
>
> [7]Mohtashami et al. Landmark Attention: Random-Access Infinite Context Length for Transformers. Adv. Neural Inf. Process. Syst. 36 (NeurIPS 2023).
>
> [8]Tworkowski et al. Focused Transformer: Contrastive Training for Context Scaling. Adv. Neural Inf. Process. Syst. 36 (NeurIPS 2023).
>
> [9]Hu et al. Efficient Length-Generalizable Attention via Causal Retrieval for Long-Context Language Modeling. Proc. Int. Conf. Mach. Learn. 42 (ICML 2025).
>
> [10]Lake et al. Generalization without Systematicity: On the Compositional Skills of Sequence-to-Sequence Recurrent Networks. Proc. 35th Int. Conf. Mach. Learn. 80, 2879 – 2888 (ICML 2018).
>
> [11]Kim et al. COGS: A Compositional Generalization Challenge Based on Semantic Interpretation. Proc. 2020 Conf. Empir. Methods Nat. Lang. Process. (EMNLP 2020), 9087 – 9105.
>
> [12]Meng et al. Locating and Editing Factual Associations in GPT. Adv. Neural Inf. Process. Syst. 35 (NeurIPS 2022).

---

> > ### Comment · Reviewer_2GPi · 2025-08-02
> >
> > I thank the authors for their thoughtful and detailed rebuttal.
> >
> > Regarding W1, I am still not entirely satisfied with the authors' response. While I now better understand the problem and acknowledge that the authors demonstrate differences in performance in real-world "large" models, I still struggle to see the relevance of the synthetic task (as presented in the rebuttal to Reviewer UvAr, Section Q1) to real-world language problems. However, I admit that I am not an expert on such synthetic tasks, so I might be setting the bar too high. Given my lingering doubts about the importance of this synthetic task, it is challenging for me to give a high recommendation for acceptance. *Nonetheless, I note here explicitly that since I am not an expert on synthetic tasks, if the Area Chair believes the task is indeed important, they may disregard this particular point.*
> >
> > Regarding W2, the paper's lack of clarity remains a significant issue. However, I acknowledge and appreciate that the authors have committed to substantially improving clarity in the manuscript.
> >
> > Considering the anticipated improvements in clarity and recognizing that another reviewer sees the synthetic task as important, I am raising my score to a 3.
> >
> > P.S. I apologize for suggesting acknowledging a paper that was published after your initial submission. However, other relevant, less recent papers exist (e.g., "Locating and editing factual associations in Mamba"), and it would be prudent to position your work clearly within the current literature. This is a minor issue and does not affect my revised score.

---

> > > ### Author Response · Authors · 2025-08-03
> > >
> > > Thank you for your thoughtful review and for investing time in our work.
> > >
> > > We share your aim of evaluating large language models through realistic tasks, though their complexity can mask a model’s core behaviors. Our preliminary experiments during the rebuttal period fell short of your expectations; we will extend this line of inquiry with more representative examples.
> > >
> > > All issues you identified will be addressed, and we will ensure that the related works include everything required. Your feedback has sharpened our perspective and will guide our revisions.
> > >
> > > We remain grateful for your guidance and look forward to improving the manuscript in light of your insights.

---

### Official Review · Reviewer_mJ4r · 2025-07-02

**Clarity:** 3
**Significance:** 3
**Originality:** 3
**Rating:** 5
**Confidence:** 3

**Summary:**

This paper studies limitations of the Mamba architecture on synthetic tasks, identifying that Mamba struggles to learn symmetrical patterns in sequential data. Through two synthetic tasks, a composite function task and an inverse sequence matching task, the authors demonstrate that Mamba exhibits a strong bias toward compositional solutions while failing to recognize symmetric patterns and relationships. The authors localize the problem to the nonlinear convolution in Mamba layers, which fuses token information in an asymmetric manner before passing it to the State Space Model (SSM) module. Through systematic analysis, they show that the limitation stems not from the SSM itself but from how token information is processed beforehand. The authors demonstrate that adding a residual connection that bypasses the convolution and writes directly to the SSM module can significantly improve performance on symmetry-requiring tasks, providing both mechanistic insights into Mamba's architectural biases and a concrete solution for improving its capabilities.

**Questions:**

- Where is the convolution operation added in the Transformer in section 4.3?

- Please move figure 5 below figure 4.

Typos:

l157: missing a space between after "Mamba"

l164: "is initializeda Gaussian" -> "is initialized as a Gaussian"

l221: "There" -> "This"

l226: "predict" -> "predicts."

l229: "network" -> "the network"

**Ethical Concerns:**

["NO or VERY MINOR ethics concerns only"]

**Final Justification:**

My original concerns on its applicability to large scale pre-trained models have been addressed in the rebuttal. I believe the paper makes a meaningful contribution, and my rating reflects my support for its publication.

**Limitations:**

yes

**Quality:**

3

**Strengths And Weaknesses:**

Strengths

- The paper provides excellent visual aids, particularly the architectural diagrams and task illustrations, which effectively communicate complex concepts.

- The composite function task and inverse sequence matching task are cleverly constructed to isolate and expose specific architectural biases, providing controlled environments that clearly reveal Mamba's symmetry-related limitations.

- The authors employ causal interventions including zero ablation (information blocking) and activation patching (information substitution) to localize the source of Mamba's limitations, providing strong causal evidence for their claims.

- The experiment modifying the convolutional weights effectively removes the asymmetric bias and demonstrates a clear causal relationship between convolution operation and the observed behavioral patterns.

Weaknesses

- The paper lacks some details necessary to understand and replicate the analysis. For instance, it is unclear how the "attention score" inside the SSM is computed for the information flow analysis, and the authors do not specify where exactly the convolution operation is added in the Transformer architecture in section 4.3.

- The study relies exclusively on synthetic tasks with relatively small models, making it unclear whether these asymmetry biases persist in large-scale pre-trained Mamba models.

- While the synthetic tasks effectively demonstrate limitations in Mamba architecture, it is unclear whether these limitations meaningfully impact Mamba's performance on language modeling or downstream tasks.

---

> ### Author Rebuttal · Authors · 2025-07-30
>
> Your feedback has been both illuminating and inspiring, and we sincerely hope to continue learning through such meaningful exchanges.
>
> **W1:**
>
> We sincerely thank you for pointing out the shortcomings in our writing. Following your valuable suggestion, we will make the following revisions:
>
> **(1)** In **Section 3.1** and the **caption of Figure 5**, we will add a clear explanation of the computation in the SSM and the definition of attention scores according to the explanation below;
>
> **(2)** In **Section 4.3** and the **Appendix**, we will include details on how to incorporate nonlinear convolution into a Transformer architecture according to the explanation below.
>
> **Attention score:** The attention score from token $j$ to token $i$ is given by the $(i, j)$ entry of the SSM matrix $S$.  due to the structural similarity between Mamba’s SSM module and the attention mechanism. Specifically, $S$ comprises both a learnable mask ($Mask$) and the term $CB^{\top}$, which is analogous to the $QK^{\top}$ component in attention.
>
> The detailed computation has been provided in the Appendix. For your convenience, we also present the full computation process of Mamba involving the SSM matrix below. Omitting trivial dimension transformations and setting the batch size to 1 to omit the batch dimension, for a given input
> $U$ to the block, the internal computation process to obtain the output
> $O$ can be described as follows:
>
> Pre-SSM
>
> $( \tilde{U}, Z, dt) = \textbf{Linear}(U),~~~ U\in R^{(s,d)}, \tilde{U}\in R^{(s,2d+2h)},Z\in R^{(s,2d)}, dt\in R^{(s,N_h)},$
>
> $(B, C, X) = \sigma(\textbf{Conv1d}(\tilde{U})),~~~ B\in R^{(s,h)}, C\in R^{(s,h)}, X\in R^{(s,2d)},$
>
> SSM
>
> $Mask = \textbf{F}(A_0, dt),~~~ A_0\in R^{N_h}, Mask\in R^{(N_h, s, s)},$
>
> $I = \textbf{Repeat}(C B^{\top},N_h)),~~~  I\in R^{(N_h, s, s)},$
>
> $S = Mask\circ I,~~~  S\in R^{(N_h, s, s)},$
>
> $Y = SX + X,~~~ Y\in R^{(N_h, s, 2d/N_h)},$
>
> Post-SSM
>
> $Y_{Norm} = \textbf{RMS}(Y\circ\sigma(Z)),~~~ Y_{Norm}\in R^{(s, 2d)},$
>
> $O = \textbf{Linear}(Y_{Norm}),~~~ O \in R^{(s, 2d)},$
>
> where
> > - $S$ : SSM matrix,
> > - $s$ : sequence length,
> > - $d$ : hidden‑state dimension,
> > - $h$ : SSM hidden dimension,
> > - $N_h$ : number of SSM heads,
> > - $\textbf{Linear}$ : linear transformation,
> > - $\textbf{Conv1d}$ : one‑dimensional convolution,
> > - $\sigma$ : nonlinear activation function,
> > - $\textbf{F}$ : function generates the $Mask$,
> > - $\textbf{Repeat}$ : dimension‑replication operation,
> > - $\circ$ : pointwise multiplication,
> > - $\textbf{RMS}$ : RMS normalization.
>
> **Transformer with convolution (which also mentioned in Question 1):** We insert a convolution after the input to the Transformer and before the attention module (applying it to $Q$, $K$, and $V$). This is analogous to how Mamba applies nonlinear convolution before the SSM module. Ignoring the batch size and number of heads for clarity, the process of computing the attention output $O$ from input $U$ can be described as follows:
>
>
> $[Q; K; V] = \textbf{Linear}(U),~~~ U\in R^{(s, d)}, Q\in R^{(s, d_q)}, K\in R^{(s,d_k)}, V\in R^{(s,d_v)}$
>
> $[Q; K; V] \leftarrow \textbf{Conv1d}[Q; K; V].$
>
> **W2 & W3:**
>
> We sincerely thank you for your valuable suggestion, which has helped us further improve our work. Following your suggestions, we conducted experiments in a practical setting using commercially deployed large language models. The results strongly support the conclusions presented in our paper, indicating that the asymmetry bias observed in Mamba persists even in large-scale models.
>
> We reformulated the inverse sequence matching task into a natural language setting,  which bears resemblance to the well-known "Passkey" (arxiv: 2305.16300), highlighting its practical significance in real-world language understanding scenarios. We performed 3 rounds of experiments on each of two models: **Mamba based** Hunyuan‑TurboS (arxiv: 2505.15431) and **Transformer based** DeepSeek-V3, with each round consisting of 100 randomly generated questions. These two models have comparable parameter scales and overall performance. The results of the experiment are as follows:
>
> | Model          | 1st. round Acc. | 2nd. round Acc. | 3rd. round Acc. | Avg. Acc. |
> | - | - | -| - | - |
> | Hunyuan‑TurboS          | 19\%     | 17\%     |17\%      |17.6\%      |
> | DeepSeek-V3 | **47\%**     | **56\%**     | **54\%**     | **52.3\%**     |
>
> The results show a clear performance gap on the inverse sequence matching task and are consistent with our earlier findings on small-scale models. This highlights that the inverse sequence matching task remains a challenge for Mamba-based models and clearly reflects the persistence of Mamba's asymmetry bias in large language models.
>
>  **Example of the task:**
>
> > **{%), )%{, %{), %){, ){%**,f...(100tokens)...Each symbol combination above is separated by commas. Identify which combination has a reverse that equals **%{)** and output that combination along with the order in which it appears. The symbol combinations are listed before the block of letters above. Perform only one match and output the result directly—do not think. You are not allowed to think step by step, just output the result directly.
>
> **Q2:**
>
> We greatly appreciate your suggestion regarding the layout, and we will revise the formatting accordingly by positioning Figure 5 below Figure 4.
>
> **C1:**
>
> **line 157:** For example, in the illustration, six tokens correspond to the pure convolutional receptive field of a two-layer Mamba($2\times3=6$).
>
> **C2:**
>
> **line 164:** A parameter $W\in R^{d_1\times d_2}$ is initialized as a Gaussian distribution $N(0,(1/d_{1}^\gamma)^2)$, where $\gamma$ is called initialization rate.
>
> **C3:**
>
> **line 221:** This raises a critical question: ...
>
> **C4:**
>
> **line 226:** ..., the network predicts the outcome with a random-guess level.
>
> **C5:**
>
> **line 229:** ..., the network can accurately predict test cases.

---

> > ### Comment · Reviewer_mJ4r · 2025-08-07
> > **Response to Rebuttal**
> >
> > Thank you for addressing my concerns and providing clarifications. The results on Deepseek and Hunyuan add strength to the overall work. I believe the paper makes a meaningful contribution, and my rating reflects my support for its publication.

---

> > > ### Author Response · Authors · 2025-08-08
> > >
> > > Thank you！

---

### Official Review · Reviewer_UvAr · 2025-07-03

**Clarity:** 3
**Significance:** 3
**Originality:** 4
**Rating:** 5
**Confidence:** 4

**Summary:**

This paper investigates the fundamental limitations of Mamba, a State Space Model (SSM) proposed as an alternate to transformers with linear complexity for long sequences. Through a series of synthetic tasks, the authors reveal that Mamba's convolution module introduces asymmetry bias. The bias impairs the recognition of symmetrical patterns, causing it to struggle with tasks requiring reversed sequence matching, and to favor compositional solutions. This work demonstrates that this limitation may stem specifically from the nonlinear convolution preceding the SSM module, rather than the SSM itself.

**Questions:**

- How would the inherent asymmetry affect language modeling tasks? What could it mean in the context of language modeling?
- What other modifications in the architecture could potentially address the asymmetry beyond residual connections?

**Ethical Concerns:**

["NO or VERY MINOR ethics concerns only"]

**Final Justification:**

I believe this work provides a simple yet effective insight that could lead to a better understanding of the Mamba architecture. While the setup in the paper could be limited, the experiments in the rebuttal verify that this analysis could potentially be extended to language modeling tasks, adding to its value.

**Limitations:**

Yes

**Quality:**

3

**Strengths And Weaknesses:**

Strengths:
- The limitation of Mamba is clearly demonstrated.
- The synthetic tasks are meticulously designed to demonstrate the limitations and inherent biases of Mamba.
- The root cause of the limitation is identified.
- Suggests a simple architectural change that leads to significant performance improvements.

Weaknesses:
- Lacks explanation on why Conv1d followed by a non-linearity inherently introduces the asymmetry, especially in Mamba architecture.

---

> ### Author Rebuttal · Authors · 2025-07-30
>
> We are deeply grateful for your recognition of our work. Your appreciation has been a great source of encouragement for us to continue pursuing this line of research, and your insightful suggestions have been invaluable in helping us improve and refine our work. We are more than willing to refine it in accordance with your evey suggestion.
>
> **W1:**
>
> We sincerely appreciate your valuable suggestions. In accordance with your advice, we will add the corresponding explanations to Section4.2 and include additional experimental results and clarifications in the appendix to elucidate why the nonlinear convolution introduces asymmetry. The following is our explanation, based on which the main text will be revised accordingly.
>
> In Mamba, sequence information is fused through a nonlinear convolution operation, which serves as the input to the SSM module. Consider two sequences, $(v_1, v_2, v_3, v_4)$ and its reversed counterpart $(v_4, v_3, v_2, v_1)$. For the convolution outputs of the above sequences, we define the final token of the result as follows:
>
> > **original sequence:** $f = c_1\circ v_1 + c_2\circ v_2 + c_3\circ v_3 + c_4\circ v_4 + \beta$
>
> > **symmetric sequence:** $g = c_1\circ v_4 + c_2\circ v_3 + c_3\circ v_2 + c_4\circ v_1 + \beta$
>
> If the convolution parameters $c_i$ are not identical, then $f$ and $g$ will generally differ. This means that even for token sequences that are symmetric in content, their representations after convolution in Mamba can be significantly different. Such a discrepancy illustrates Mamba’s inherent asymmetry, as it fails to preserve the equivalence of symmetric inputs.
>
> The root cause of this asymmetry lies in the non-uniformity of the convolution weights. Since the convolution parameters in Mamba are initialized randomly and trained independently, the $c_i$ values typically remain distinct throughout training. As a result, the convolution operation induces a persistent asymmetry, where different token orders lead to different outputs. We examined the cosine similarity between the individual parameters of the convolution kernel at both the beginning and end of training, and found that they were largely orthogonal to one another, indicating a strong and persistent inconsistency throughout the training process.
>
> Moreover, as shown in the main text, when we manually set every  $c_i$ to an all‑ones vector, the differences vanish; the convolution becomes insensitive to token order, and $f$ equals $g$. Once the convolution ceases to introduce asymmetry, as shown in the experiment, Mamba—under the same experimental settings—no longer favors the composite solution but instead learns the symmetric solution.
>
> The detailed convolution computation process in Mamba, for an input $V$, to obtain the convolutional output $O$ can be expressed as follows:
>
> $V = (v_1, v_2, ..., v_n),~~~ V\in R^{(n,m)}, v_i\in R^m,$
>
> $C = (c_1, c_2, c_3, c_4),~~~ C\in R^{(4,m)}, c_i\in R^m,$
>
> $p_i = c_1\circ v_{i-3} + c_2\circ v_{i-2} + c_3\circ v_{i-1} + c_4\circ v_{i},~~~ p_i\in R^m,$
>
> $o_i = p_i + \beta,~~~ o_i\in R^m, \beta\in R^m$
>
> $v_j = \vec{0}\in R^{m},~~~ j \leq 0,$
>
> $O = \textbf{Conv1d}(V) = (o_1, o_2, ..., o_n),~~~ O\in R^{(n,m)}$
>
> Where $C$ denotes the convolution kernel parameters; $\beta$ is the convolution bias. The four vectors of $C$ are multiplied point‑wise with the corresponding $v_i$ at each position. When the index of $v_i$ is smaller than 1, $v_i$ is set to 0 as padding.
>
> **Q1:**
>
> We are truly grateful for your question, which has provided valuable guidance for improving our work.
>
> Symmetric relationships are common in real-world tasks—for example, the rotational invariance implied by the word and in natural language. To investigate how Mamba’s inherent asymmetry affects language tasks, we concentrate on tasks that fundamentally rely on symmetric relational structures.
>
> In order to respond to your inquiry, we converted the inverse sequence matching task into a natural‑language format and carried out additional experiments, thereby providing further evidence for the conclusions in our work. Note that, this task is very similar to a widely used task "Passkey" (arxiv: 2305.16300). The experimental results reveal the impact of Mamba’s inherent asymmetry on its performance in language modeling tasks. We first present the main results and findings; a complete task example and a detailed explanation follow below.
>
> To observe the effect of Mamba’s inherent asymmetry in language modeling with truly practical setting, we conducted experiments on two models: Hunyuan‑TurboS (arxiv: 2505.15431), a well-known large language model with 57 Mamba layers, and DeepSeek-V3, a Transformer-based model with comparable parameter scale and performance. The details and results of the experiment are as follows:
> - **3** rounds of testing, each comprising **100** randomly generated questions.
> - **Average accuracy:** Hunyuan TurboS = **17.6\%** (19\%, 17\%, 17\%), DeepSeek-V3 = **52.3\%** (47\%, 56\%, 54\%).
>
> These results, obtained in a real-world application scenario with large-scale models and natural language inputs, are consistent with the findings from our paper based on small models and simplified data, thus further validating our conclusions. This demonstrates that the asymmetry bias inherent to Mamba, as highlighted in our work, not only significantly impacts small models but remains evident in large language models as well. It suggests that this asymmetry makes Mamba less suitable for natural language tasks that rely heavily on symmetric relations.
>
> Example of the Passkey task used in our test:
>
> > **{%), )%{, %{), %){, ){%**,f...(100tokens)...Each symbol combination above is separated by commas. Identify which combination has a reverse that equals **%{)** and output that combination along with the order in which it appears. The symbol combinations are listed before the block of letters above. Perform only one match and output the result directly—do not think. You are not allowed to think step by step, just output the result directly.
>
> **Remarks**:
> - We replaced the numeric sequences in the original task with combinations of symbols to avoid the influence of tokenization effects.
> - The task input consists of five uniquely ordered symbol combinations followed by 100 randomly generated letters, which serve to prevent Mamba from accessing information directly via convolution.
> - The final prompt is designed to clearly convey the task objective to the model while discouraging it from relying on chain-of-thought reasoning, in order to more closely replicate the conditions under which the small-scale models were tested in our paper.
>
> **Q2**
>
> We sincerely appreciate your question, which has led us to a deeper investigation and motivated us to conduct further experiments. The inherent asymmetry of Mamba arises from the asymmetry of its convolution operation. To mitigate this issue, one can introduce additional channels for information flow, thus avoiding the original design of Mamba that fuses all information solely through nonlinear convolution.
>
> Residual connections seem to be the most straightforward solution to introduce additional channels. Such connections allow the SSM in Mamba to access both the asymmetrically fused token representations and the original, unfused token information, enabling the model to adaptively choose the appropriate source based on the task. This could help mitigate the limitations caused by the inherent asymmetry, while still preserving the unique characteristics of Mamba.
>
> In addition, inspired by the original Mamba architecture, another possible way to inject the raw token information is through a "gating mechanism". For example, one might use a pointwise product between the original token and the fused token (after nonlinear convolution) as the input to the SSM. We additionally conducted multiple experiments on this approach, and found that it also enables Mamba to achieve 100\% accuracy on the test set of the reverse sequence matching task. However, its out-of-distribution (OOD) accuracy is slightly lower than that achieved with residual connections. Under standard initialization, the results are as follows:
>
> | Model          | Train Acc. | Test Acc. | **OOD Acc.** |
> | - | - | -| - |
> | Mamba          | 100.0%     | 20.7%     |**19.6%**      |
> | Residual-Mamba | 100.0%     | 100.0%     | **63.6\%**     |
> | Gating-Mamba | 100.0%     | 100.0%    | **42.9\%**     |
>
> and the setup is identical to that described in Figure 9 of the original paper.
>
> This may be due to the fact that, compared to residual connections, this method leads to a deeper entanglement between the original token information and the fused representation produced by the nonlinear convolution. In contrast, residual connections simply add the original and fused token representations. If these two types of information reside in different subspaces, the additive structure still allows for the preservation and independent extraction of each type of information.
>
> The following presents the computation process of the gating mechanism. Given the input $U$ to the Mamba block, we have:
>
> $( \tilde{U}, Z, dt) = \textbf{Linear}(U),~~~ U\in R^{(s,d)}, \tilde{U}\in R^{(s,2d+2h)},Z\in R^{(s,2d)}, dt\in R^{(s,N_h)},$
>
> $(B, C, X) = \sigma(\textbf{Conv1d}(\tilde{U})),~~~ B\in R^{(s,h)}, C\in R^{(s,h)}, X\in R^{(s,2d)},$
>
> $(\tilde{B}, \tilde{C}, \tilde{X}) = (B, C, X)\circ \tilde{U},~~~ \tilde{B}\in R^{(s,h)}, \tilde{C}\in R^{(s,h)}, \tilde{X}\in R^{(s,2d)},$
>
> Finally, $\tilde{B}$, $\tilde{C}$, and $\tilde{X}$ serve as the input to the SSM.

---

> > ### Comment · Reviewer_UvAr · 2025-08-06
> >
> > Thank you for the detailed follow up. The mathematical formulation is very intuitive, and would be a valuable addition to the revised draft for extra clarity.
> >
> > The questions I raised were were less about what needs to be addressed in the main paper, because I believe the work was already solid, so I would like to thank the authors for designing and conducting experiments. The experiments match the paper's intuition/claim, demonstrating that their results and insights could lead to more interesting works in the future.

---

> > > ### Author Response · Authors · 2025-08-07
> > >
> > > Thank you for your valuable insights in elevating our work and we will continue to deepen our research.

---

### Note · Authors · 2025-08-14

We are deeply grateful for the reviewers’ guidance. We have taken the suggestions seriously, revised the paper accordingly, and added further experiments.

Our study isolates a fundamental difference between Mamba and Transformer—Mamba’s asymmetry bias，and through extensive evidence, traces this bias to Mamba’s use of nonlinear convolution. Below we explain why we adopted this methodology and why it matters.

## Why use synthetic data? ##

Our goal is to examine how architectural differences (nonlinear convolution in Mamba) lead to distinct behaviors, manifested as preferences and performance on certain tasks.

Real world tasks are highly complex, making controlled ablations and mechanism observation difficult. We therefore distill such tasks into simple, analyzable synthetic tasks (e.g., composite function task). This lets us study model properties cleanly, without the confounds of task complexity.

With synthetic data, we can probe internal information flow, and run sharply targeted controls (e.g., information flow blocking) to reveal properties and their causes. Once a property and its structural source are identified, we can validate with tasks tailored to the hypothesis (e.g., inverse sequence matching to test symmetry), showing that Mamba struggles with symmetric relations. Each step relies on the low complexity of synthetic setups to make the analysis transparent and hence our choice.

## Do the findings generalize? ##

Synthetic tasks clarify properties and their causes, generalization is established if those properties persist in larger models and real settings. We therefore converted the inverse sequence matching task into a natural language version and tested it on commercial LLMs. The results mirrored our small model findings: although Mamba-based LLM (Hunyuan TurboS = 17.6%) perform comparably to Transformer-based LLM (DeepSeek-V3 = 52.3%) on many benchmarks, it performs significantly worse on this task than its counterpart. This both supports generalization and underscores the value of synthetic analysis: synthetic tasks reveal model's characteristics (Mamba’s asymmetry bias) that overall benchmarks often hide.

We believe research grounded in synthetic data can deepen the field’s understanding of model behavior and ultimately inform architectural improvements in large language models.

Finally, we thank the reviewers once again for their time and constructive feedback. The discussion has been invaluable in strengthening our work.

---

### Decision · Program_Chairs · 2025-09-17

**Decision:**

Accept (spotlight)

**Comment:**

The paper probes architectural differences between Mamba (SSM-based model) and Transformers using controlled synthetic tasks—composite function learning and inverse sequence matching. By experiments, the authors have interesting findings revealing fundamental differences between Mamba and Transformer models, particularly in handling symmetrical patterns and relationships. They propose simple mitigations (e.g., residual paths / gating around the convolution) and shows good potential of improving SSM models.

In rebuttal, the authors also add large-model experiments (e.g., Hunyuan-TurboS vs. DeepSeek-V3) indicating the asymmetry effect persists in scaled settings (17.6% vs. 52.3% on the Passkey task).

Strengths
Multiple reviewers highlight that the paper convincingly traces the failure mode to the nonlinear convolution via targeted interventions and weight modifications.
The synthetic tasks are minimal but effective to test compositional vs. symmetric solutions.
Proposed residual/gating fixes are simple but useful

Weaknesses
The only concern after rebuttal is "Why the bias is bad”. But the authors give good explanation by experimenting on large models.